# Tumor suppressor mediated ubiquitylation of hnRNPK is a barrier to oncogenic translation

Bartosz Mucha[1], Shuo Qie [1], Sagar Bajpai[1], Vincenzo Tarallo [1], J. Nathaniel Diehl[2], Frank Tedeschi[1,3], Gao Zhou[3], Zhaofeng Gao[4], Samuel Flashner [5], Andres J. Klein-Szanto [6], Hanina Hibshoosh[5], Shimonosono Masataka[5], Olga S. Chajewski[7], Ireneusz Majsterek [8], Dariusz Pytel[7,8], Maria Hatzoglou [4], Channing J. Der[2,9,10], Hiroshi Nakagawa[11], Adam J. Bass [5], Kwok-Kin Wong [12], Serge Y. Fuchs [13], Anil K. Rustgi[11], Eckhard Jankowsky[1,3,14] & J. Alan Diehl [1,3,14] ✉

Heterogeneous Nuclear Ribonucleoprotein K (hnRNPK) is a multifunctional RNA binding protein (RBP) localized in the nucleus and the cytoplasm. Abnormal cytoplasmic enrichment observed in solid tumors often correlates with poor clinical outcome. The mechanism of cytoplasmic redistribution and ensuing functional role of cytoplasmic hnRNPK remain unclear. Here we demonstrate that the SCF^Fbxo4 E3 ubiquitin ligase restricts the pro-oncogenic activity of hnRNPK via K63 linked polyubiquitylation, thus limiting its ability to bind target mRNA. We identify SCF^Fbxo4-hnRNPK responsive mRNAs whose products regulate cellular processes including proliferation, migration, and invasion. Loss of SCF^Fbxo4 leads to enhanced cell invasion, migration, and tumor metastasis. C-Myc was identified as one target of SCF^Fbxo4-hnRNPK. Fbxo4 loss triggers hnRNPK-dependent increase in c-Myc translation, thereby contributing to tumorigenesis. Increased c-Myc positions SCF^Fbxo4-hnRNPK dysregulated cancers for potential therapeutic interventions that target c-Myc-dependence. This work demonstrates an essential role for limiting cytoplasmic hnRNPK function in order to maintain translational and cellular homeostasis.

Most cellular RNAs associate with RNA Binding Proteins (RBPs) within ribonucleoprotein complexes. RBPs influence RNA structure and play critical roles in RNA biogenesis, stability, function, transport, and cellular localization[1,2]. Alterations in expression or mutation in either RBPs or their RNA targets is associated with human diseases, including muscular atrophies, neurological disorders, and cancer[3]. HnRNPK, the founding member of the KH-domain containing family of RBPs, is modestly overexpressed in different types of tumors such as esophageal and head and neck squamous cell carcinomas and melanoma[4–6]. Notably, increased localization of hnRNPK to the cytoplasm in cancers has been associated with poor prognosis, suggesting that cytoplasmic functions of hnRNPK may be tumorigenic, thus highlighting an unappreciated potential layer of hnRNPK regulation[6–12].

Protein ubiquitylation plays a critical role in cell growth and survival. While the proteolytic roles of ubiquitylation are widely appreciated, ubiquitylation can also function as a signal to alter protein activity, localization, and protein–protein interactions. Ubiquitin chains are formed via utilization of internal lysines and differential utilization of these lysines determines outcome. Polyubiquitin chains formed through lysine 48 generally signals destruction through the 26S proteasome, while chains utilizing lysine 63 typically regulate protein interactions and function[13,14].

The SCF (Skp-Cullin 1-Fbox) E3 ligases regulate a broad set of growth regulatory proteins. Members of the F-box family of proteins act as adaptors to bridge ligase core components with the substrate[15]. The salient feature of these proteins is the F-box motif (located near the N-terminus), which is necessary for binding Skp1; Skp1 in turn, associates with the Cul1-Rbx1 complex, which recruits E2 ubiquitin conjugating enzymes[16]. SCF E3 ligases regulate a variety of targets implicated as drivers or suppressors of cancer. For example, Fbxo4 (SCF^Fbxo4) regulates the degradation of the cyclin D1 oncoprotein and loss of Fbxo4 in mice results in a variety of cancers that are highly metastatic[17]. Tumor suppressive functions of Fbxo4 are only partially cyclin D1-dependent[18,19], highlighting the existence of key unidentified, pro-tumorigenic targets. The current investigation reveals the role for SCF^Fbxo4 in the regulation of mRNA translation through ubiquitylation of hnRNPK. This investigation highlights the role for a SCF^Fbxo4-hnRNPK pathway that determines cell proliferation and metastasis through regulation of c-Myc protein synthesis and highlights potential therapeutic opportunities associated with cancers harboring loss of Fbxo4.

## Results

### Polyubiquitylation of hnRNPK by the SCF^Fbxo4 ubiquitin ligase

To evaluate the role of SCF^Fbxo4 in shaping transcriptional and translational landscapes, we conducted transcriptional profiling and Ribo-Seq on mRNA from Fbxo4^+/+ (WT) and Fbxo4^−/− MEFs (Fig. 1A). Applying a twofold change cut-off, we identified 3649 mRNAs dysregulated upon Fbxo4 loss (Fig. 1B). Since Fbxo4 is an E3 ubiquitin ligase component, we reasoned that regulation of mRNAs was indirect and could result from differential regulation of RBPs by SCF^Fbxo4. Hence, we performed RNA motif enrichment using oRNAment algorithm[20] and noted enrichment in sequences recognized by KH-domain family RBPs (Fig. 1C, D). FXR1, a KH-domain RBP family member, was previously identified as a SCF^Fbxo4 substrate suggesting a potential SCF^Fbxo4 target specificity toward KH domain proteins[21]. Analyzing material collected during a previous purification/mass spectrometry experiment in which FXR1 was identified as a Fbxo4 target[21], we noted multiple KH-domain RBPs that co-purified with Fbxo4 with hnRNPK as one of high potential relevance (Fig. 1E). We noted that the mRNAs (RNA and RPF components) dysregulated upon Fbxo4 ablation were highly enriched in hnRNPK motifs (Fig. 1F). Of note, oRNAment analysis on targets responding at the level of RPF have also been run and show the same (3rd) position of hnRNPK in terms of motif abundance within KH domain RBPs (data not shown). We therefore assessed the number of Fbxo4-responsive targets that were also hnRNPK-dependent. A CLIP-seq bioinformatic data base in Encyclopedia of RNA interactome database (Encori, starBase2−https://starbase.sysu.edu.cn/)[22] was used to generate a list of hnRNPK mRNA targets to compare with mRNAs identified as >2-fold up or down-regulated in the Fbxo4^+/+ versus Fbxo4^−/− MEFs from either RNA-seq or Ribo-seq. Nearly 50% (1722 included in analysis) of the Fbxo4 responsive genes were shared with the collection of hnRNPK targets, highlighting major overlap with targets altered at both the mRNA level relative to mRNAs protected by ribosomes (Fig. 1G).

Based upon these observations, we postulated that hnRNPK is regulated by SCF^Fbxo4. Consistently, endogenous (Figs. 1H and S1A) and ectopic Fbxo4 (wt and catalytically deficient ΔF) coprecipitated with hnRNPK (Fig. S1B). Mutational analysis of Fbxo4 revealed that hnRNPK binds to the C-terminal substrate binding domain of Fbxo4 (Fig. S1C, D).

A majority of known hnRNPK protein-protein interactions are mediated by its KI domain (Fig. S1C)[23]. Consistently, deletion of the KI domain of hnRNPK inhibited Fbxo4-hnRNPK binding (Fig. S1E).

### SCF^Fbxo4 catalyzes K63 linked polyubiquitylation of hnRNPK

The data above suggest a potential substrate-E3 ligase relationship. Consistently, overexpression of wt Fbxo4 but not the catalytically inactive mutant Fbxo4^ΔF catalyzed strong polyubiquitylation of hnRNPK (Fig. 2A). Analogous to previously identified substrates such as cyclin D1, SCF^Fbxo4 required αB-Crystallin as a co-factor for polyubiquitylation of hnRNPK (Figs. 2A and S2A)[24]. Inhibition of neddylation with MLN4924[25], abrogated hnRNPK polyubiquitylation (Fig. 2A) further supporting a role for SCF^Fbxo4 as an E3 ligase targeting hnRNPK. Finally, purified SCF^Fbxo4 but not SCF^Fbxo4ΔF (deletion of F-box inhibits recruitment of core E3 ligase components) catalyzed hnRNPK ubiquitylation in vitro (Figs. 2A, B and S2B).

Previous work has demonstrated SCF^Fbxo4 catalyzes polyubiquitin chains linked through lysine 48 (K48); K48 linked polyubiquitin of cyclin D1 and Fxr1 trigger proteasomal degradation[21,26]. Surprisingly, depletion of Fbxo4 in multiple cell lines affected neither hnRNPK protein levels (Fig. S2C) nor its half-life (Fig. S2D) suggesting SCF^Fbxo4 ubiquitylation of hnRNPK is not a signal for degradation. Proteosome inhibition did not increase binding of endogenous hnRNPK and Fbxo4 as would be expected in case of degradation control (Fig. S2E). Thus, we focused on the second most abundant type of polyubiquitin linkage, elongation through lysine-63 (K63). Indeed, SCF^Fbxo4 preferentially catalyzed K63-linked polyubiquitin chains on purified hnRNPK (Fig. 2C, D). In contrast, SCF^Fbxo4 preferentially catalyzed K48-linked polyubiquitin chains on Fxr1[21].

Subsequent efforts focused on identification of structural features that mediate Fbxo4-hnRNPK regulation. Amino acid sequence alignment revealed a short fragment of hnRNPK with high homology with the cyclin D1 degron (Fig. 2E). This region of hnRNPK overlaps with one of two nuclear localization signals (NLS) and is flanked by two lysines at position 21 (K21) and 34 (K34). Site-directed mutagenesis of K21 to arginine (K21R) led to partial loss of ubiquitylation while complete loss was observed in the K34R mutant and in the double mutant (K21/34R) (Figs. 2F and S2F). While in vitro ubiquitylation did not reveal a robust loss of signal in the K34R mutant (Fig. S2G), truncation of the hnRNPK N-terminal region (Fig. S1C) resulted in complete lack of ubiquitylation signal (Fig. S2H) consistent with our interpretation that the differential pattern of signal loss reflects differences in the nature of the in vivo versus in vitro experiments. Consistent with SCF^Fbxo4 utilizing K63 ubiquitin linkages for hnRNPK, expression of ubiquitin K63R, but not ubiquitin K48R, abolished SCF^Fbxo4-dependent polyubiquitylation of hnRNPK (Fig. 2G). Accordingly, purified SCF^Fbxo4 preferentially utilized ubiquitin mutant harboring only K63 for hnRNPK (Fig. 2C), while K48 was utilized preferentially for cyclin D1 (Fig. S2I).

In benign cells, hnRNPK is predominantly nuclear implying that cytoplasmic function is restricted[27,28]. By contrast in cancer cells, hnRNPK accumulates in the cytoplasm arguing for gain of cytoplasmic function in cancer[6–12]. SCF^Fbxo4 facilitates the proteasomal degradation of all its known substrates mainly in the cytoplasm[29]. Therefore, we addressed whether Fbxo4-dependent ubiquitylation may define hnRNPK trafficking. Fractionation of cells showed no changes in hnRNPK localization for either wild type or ubiquitin-refractory hnRNPK mutant (Fig. S2J). In vivo ubiquitylation assay showed presence of ubiquitylated hnRNPK only in cytoplasm (Fig. 2H). These findings suggest SCF^Fbxo4 regulates hnRNPK cytoplasmic function but does not alter hnRNPK localization.

### SCF^Fbxo4-dependent hnRNPK polyubiquitylation determines the rate of c-Myc protein synthesis via 5′UTR initiated translation and ribosome accumulation at ORF

To address the biological significance of Fbxo4-mediated regulation of hnRNPK, we used Gene Set Enrichment Analysis (GSEA) to identify signatures associated with Fbxo4 loss. A c-Myc signature was associated with transcripts both up- and down-regulated (Figs. 3A and S3A). A significant increase of c-Myc protein was observed Fbxo4^−/− MEFs (Figs. 3B and S3B), while c-Myc mRNA was unaffected (Fig. S3C). The increase in c-Myc protein was rescued by knock-down of hnRNPK

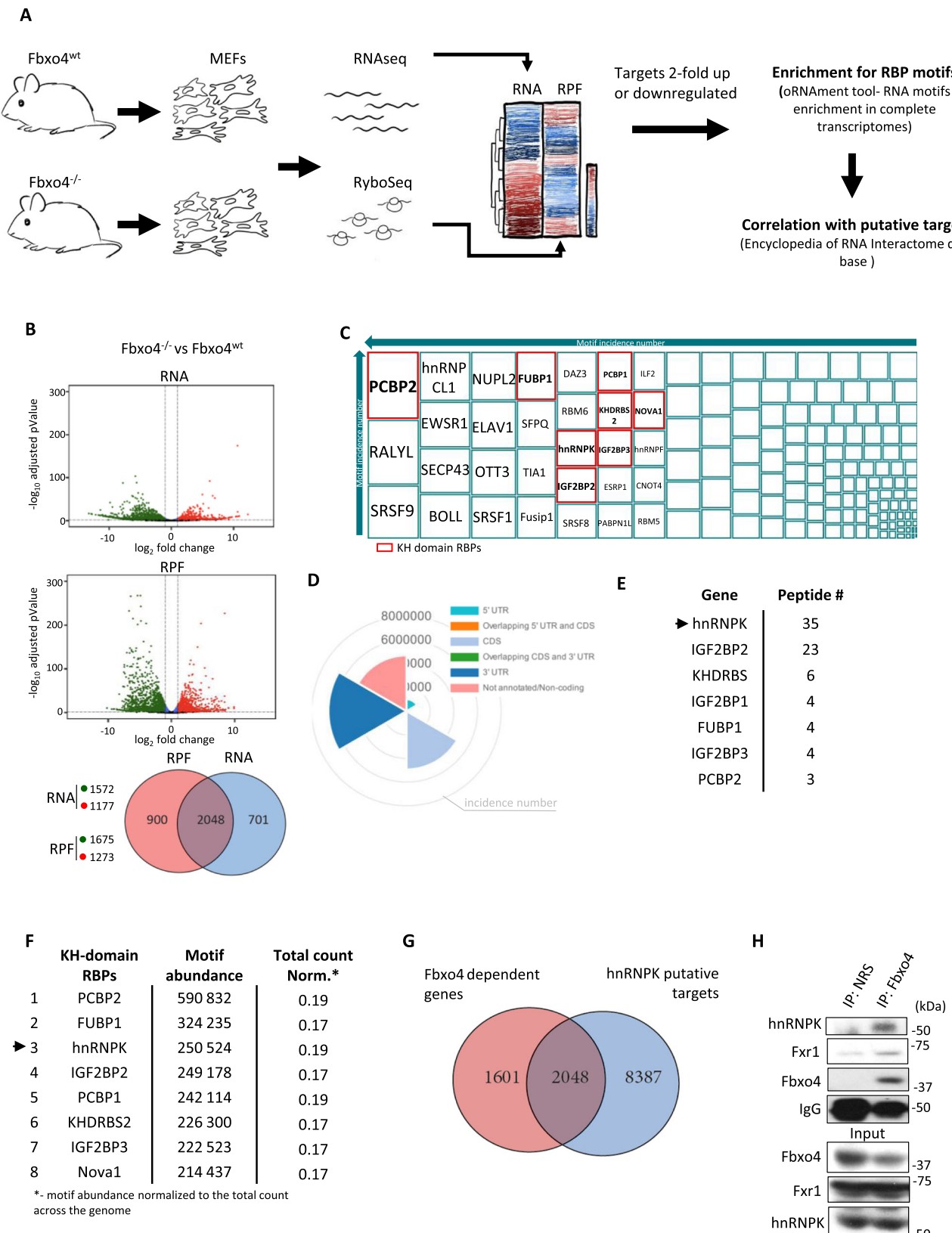

(Fig. 3B) or re-introduction of Fbxo4$^{wt}$ (Fig. 3C). Overexpression of ubiquitin-refractory hnRNPK mutants but not hnRNPK$^{wt}$ increased c-Myc (Fig. S3D, E) demonstrating that both Fbxo4 and hnRNPK contribute to c-Myc regulation.

To further evaluate the hnRNPK-c-Myc relationship and identify other potential SCF$^{Fbxo4}$-hnRNPK downstream effectors, we employed ribosome profiling. The tumor suppressive properties of SCF$^{Fbxo4}$ and oncogenic activity of hnRNPK dysregulation, suggest that SCF$^{Fbxo4}$ ubiquitylation in the cytoplasm inhibits hnRNPK cytoplasmic functions. Applying this logic, we performed ribosome profiling on all combinations of hnRNPK/Fbxo4 depletion. Sequencing of total RNA and RPFs (ribosome protected fragments) was performed on MEFs

**Fig. 1 | hnRNPK is a SCF$^{Fbxo4}$ target. A** The flow of genomic analysis. **B** Volcano plots representing genes differentially regulated upon Fbxo4 knock-out at the total mRNA level (RNA) and mRNA protected by ribosomes (RPF), *p* values were generated by DESeq2 applying the Wald test followed by Benjamini and Hochberg method correction; genes with *p* < 0.05 and 2<fold upregulated are represented by red dots, genes with *p* < 0.05 and 2<fold downregulated are represented by green dots, genes with *p* < 0.05 and 2 >fold up or downregulated represented by blue dots, all remaining genes are represented by black dots. **C** Genes up or down regulated by twofold change and adj. *p* value < 0.05 (identified in **B**) in Fbxo4-deficient MEFs were subjected to analysis of putative RBP-binding site instances with the use of oRNAment tool, **D** the localization of identified motifs is presented on the wheel graph. **E** Peptide recovery from mass-spec of RBPs coprecipitated with Fbxo4 performed previously[21]. **F** summary of oRNAment analysis top scores of KH-domain RBPs. **G** Fbxo4-dependent RNA and RPF targets were compared with the hnRNPK target reported in the Encyclopedia of RNA Interactome data base (Encori, https://starbase.sysu.edu.cn/; data parameters-–RBP=hnRNPK, genome=human, assembly=GRCh37-hg19, stringency=2). **H** Endogenous hnRNPK co-purifies with Fbxo4 in HEK293T cells, Fxr1 is presented as a positive control.

under the following conditions: wt- control (I), Fbxo4 knock-out (II), Fbxo4 knock-out + hnRNPK RNAi knock-down (III) and RNAi hnRNPK knock-down only (IV) (Fig. S3F). The data was assessed as fold change values for RNA versus RPF (Fig. 3D). To identify genes controlled by SCF$^{Fbxo4}$-hnRNPK, we postulated that hnRNPK is capable of either enhancement or inhibition of mRNA levels, translation or both. We established selection criteria for each condition to identify SCF$^{Fbxo4}$-hnRNPK dependent targets. We included all the genes where RNA and/or RPF parameters are up or down-regulated at least two-fold upon the loss of Fbxo4. These targets were rescued by at least 25% when hnRNPK was knocked down in Fbxo4$^{-/-}$ cells. Finally, we expected that depletion of hnRNPK should have the opposite impact as Fbxo4 loss; approximately 700 SCF$^{Fbxo4}$-hnRNPK downstream targets were identified by these criteria. Among these, 77 are c-Myc targets based on GSEA datasets (Fig. 3E).

Since c-Myc is an unstable protein, we felt it imperative to assess degradation. No binding of Fbxo4 with c-Myc was noted in (Fig. S3G). Additionally, no change in the rate of c-Myc degradation was observed following disruption of SCF$^{Fbxo4}$ (Fig. S3H) implying no direct impact on c-Myc ubiquitin-dependent degradation.

The increase of c-Myc RPFs in Fbxo4$^{-/-}$ cells corresponds with observed higher protein output. Upon hnRNPK knock-down, the apparent translation efficiency (RPF/RNA ratio) increased (Fig. 3F), while c-Myc protein level was reduced (Fig. 3B). To understand this paradox, we analyzed the distribution of RPF reads across the c-Myc ORF. The higher apparent translation efficiency in Fbxo4$^{-/-}$ cells indicates increased ribosomal occupancy and correlates with a robust increase in c-Myc protein synthesis upon Fbxo4 loss (Fig. S3I). Interestingly, RPF distribution following hnRNPK loss revealed a transcript region where the RPF occupancy is elevated regardless of Fbxo4 status (Fig. 3G; red arrows). These peaks identify sites of ribosome accumulation in the absence of hnRNPK. Since c-Myc synthesis declined, the peaks noted are consistent with ribosome pausing at these sights as a cause for the noted reduction in c-Myc levels and synthesis. From these data, we concluded that SCF$^{Fbxo4}$-dependent regulation of hnRNPK activity contributes to ribosome accumulation at these sites.

To address the mechanism, we considered that SCF$^{Fbxo4}$ might regulate hnRNPK-dependent c-Myc protein synthesis at its Internal Ribosome Entry Site (IRES). A bi-cistronic reporter system with cap-dependent regulated *Renilla* luciferase (an internal control), and c-Myc IRES sequence upstream of the *Firefly* luciferase was used to evaluate how hnRNPK ubiquitylation affects c-Myc translation (Fig. 3H). Expression of hnRNPK$^{wt}$ stimulated c-Myc IRES by 50% while the hnRNPK$^{K21/34A}$ ubiquitylation-deficient mutant was more potent in promoting c-Myc IRES activity (Fig. 3I). Conversely, knock-down of hnRNPK significantly silenced c-Myc IRES but had no effect on the control hairpin construct (Fig. S3J). In addition, co-overexpression of Fbxo4 with hnRNPK$^{wt}$ but not hnRNPK$^{K21/34A}$ inhibited hnRNPK-dependent c-Myc IRES stimulation (Fig. 3J). Pulse-labeling of total protein revealed no changes in global protein synthesis (Fig. S3K) consistent with hnRNPK-target specificity. Finally, to directly address whether SCF$^{Fbxo4}$ regulates c-Myc protein synthesis, protein synthesis dynamics were tracked by cycloheximide-induced depletion of c-Myc and subsequent release with addition of proteasome inhibitor MG132 to prevent degradation of de novo synthesized c-Myc. This analysis revealed a significant increase in c-Myc synthesis in Fbxo4$^{-/-}$ versus Fbxo4$^{+/+}$ cells (Fig. 3K). Further, through a combination of in vivo ubiquitylation assays and protein labeling through incorporation of the methionine analog L-azidohomoalanine (AHA) followed by click chemistry to tag AHA with TAMRA dye, we found that the poly-ubiquitylation status of hnRNPK correlates with delayed c-Myc synthesis (Fig. 3L).

To assess if c-Myc is a direct hnRNPK target that is in turn regulated by SCF$^{Fbxo4}$, we performed RNA immunoprecipitation (RIP). RIP analysis highlighted enrichment of c-Myc mRNA in the ubiquitylation-refractory hnRNPK mutants (Fig. 3M). Additionally, SCF$^{Fbxo4}$-dependent ubiquitylation of hnRNPK prior to UV crosslinking of protein:mRNA complexes inhibited the ability of hnRNPK to precipitate bulk RNA consistent with a model wherein ubiquitylation of hnRNPK reduces its ability to associate with RNA (Fig. S3L).

Overall, the biochemical role of hnRNPK ubiquitylation can be defined by the following crucial observations: (1) ubiquitylation lowers hnRNPK's affinity to RNA (Fig. S3L) including c-Myc mRNA (Fig. 3M); (2) ubiquitylation-refractory hnRNPK mutant stimulates c-Myc translation through 5'UTR IRES (Fig. 3I, J); (3) ubiquitylation status of hnRNPK correlates with a lower c-Myc synthesis rate (Fig. 3L). Together these observations demonstrate that according to the expected tumor suppressor nature of Fbxo4, SCF$^{Fbxo4}$ functions as a limiting factor for hnRNPK-mediated oncogenic translation.

## SCF$^{Fbxo4}$-hnRNPK-c-Myc regulates cell proliferation

To assess the impact of SCF$^{Fbxo4}$-hnRNPK regulation on cell proliferation, we compared Fbxo4$^{+/+}$ versus Fbxo4$^{-/-}$ MEFs following over-expression of hnRNPK. Basal doubling time of Fbxo4$^{-/-}$ is shorter than Fbxo4$^{+/+}$ MEFs (Fig. S3M) consistent with increased levels of cyclin D1[19,24,26] and c-Myc. Expression of hnRNPK$^{K21/34A}$ mutant in Fbxo4$^{+/+}$ MEFs increased proliferation to a greater extent than hnRNPK$^{wt}$ reflecting loss of ubiquitin-dependent control of hnRNPK$^{K21/34A}$. In Fbxo4$^{-/-}$ MEFs, hnRNPK$^{K21/34A}$ and hnRNPK$^{wt}$ promoted faster proliferation equally, consistent with loss of SCF$^{Fbxo4}$-dependent action on hnRNPK$^{wt}$ in these cells (Fig. S3M). Importantly, knockdown of hnRNPK prevented proliferation regardless of Fbxo4 status (Fig. S3N).

## SCF$^{Fbxo4}$-hnRNPK regulates cell invasion and migration

To predict the potential phenotypic output of SCF$^{Fbxo4}$-hnRNPK differentially regulated genes, all 693 putative targets identified in Fig. 3E were used as an input in the pathway enrichment analyses using KEGG, Reactome and Gene Ontology-biological process (GO_BP). Reactome and KEGG highlighted significant changes for genes involved in cell migration, adhesion and angiogenesis, all processes involved in cancer progression (Figs. 4A, B and S4A). We noted enrichment for Gene Ontology-cellular compartment (GO_CC), where highest probability was represented by genes which protein products belong to cellular membrane and extracellular space categories (Fig. S4B).

Since Fbxo4 deficiency triggers an enrichment of pathways associated with cell adhesion, motility, and microenvironment rearrangement, we evaluated SCF$^{Fbxo4}$-hnRNPK regulation of cell invasion and motility. Cells were subjected to chemotaxis-dependent invasion assay through extracellular matrix (ECM). To eliminate the impact of hnRNPK/Fbxo4 dependent cell viability regulation (Fig. S3M, N) on

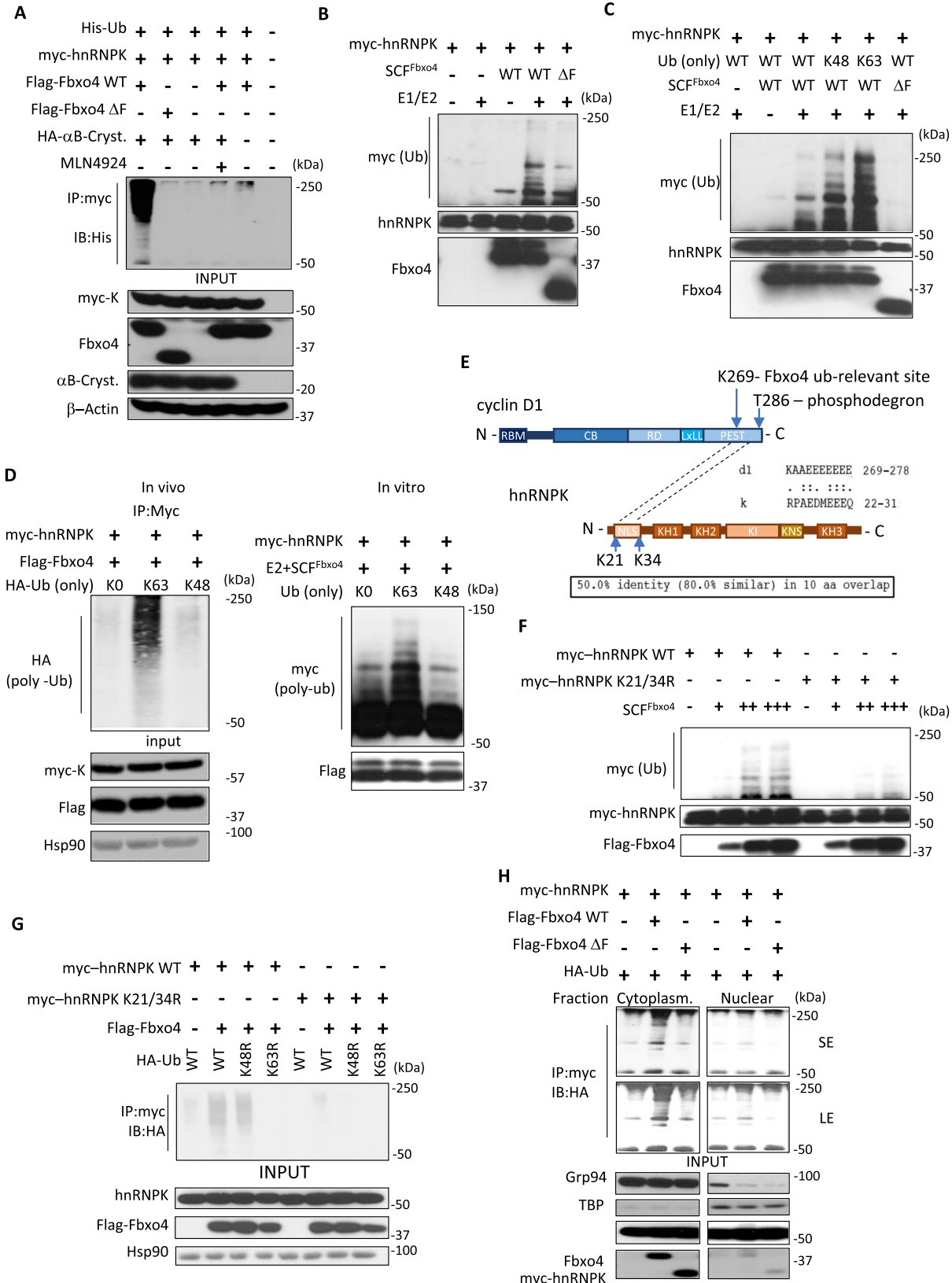

invading cell number, all cells were serum-deprived to prevent their proliferation. Fbxo4$^{-/-}$ MEFs were significantly more invasive than wild type MEFs; RNAi-mediated depletion of hnRNPK reduced invasiveness (Fig. 4C, D). Intriguingly, invasiveness was increased by expression of the ubiquitylation-deficient hnRNPK$^{K21/31A}$ mutant but not by expression of hnRNPK$^{wt}$ (Fig. S4D) highlighting a role for hnRNPK in driving

this phenotype. C-Myc depletion reduced invasion by >50%, an impact similar to that observed with hnRNPK depletion (Fig. 4E, F). As an independent approach of evaluating the role of c-Myc, we utilized JQ1, a small molecule regulator of bromo-domain factors and an established agent capable of depleting c-Myc[30]. JQ1 treatment significantly abrogated c-Myc expression and cell invasion without impacting

**Fig. 2 | SCF$^{Fbxo4}$ catalyzes hnRNPK ubiquitylation in vivo and in vitro. A** In vivo and **B** in vitro polyubiquitylation of hnRNPK is achieved by full-length Fbxo4 but not catalytically deficient Fbxo4 with the deletion of F-box cassette. **C** In vitro ubiquitylation reaction with the use ubiquitin molecules retaining K48 or K63 only indicates dominant presence of K-63 linked polyubiquitin chains. **D** In vivo (performed in HEK293T cells) and in vitro ubiquitylation assay with single lysine retaining ubiquitin mutants (K63 or K48) or no-lysine mutant K0 as a background control. **E** Amino acid sequence alignment between known Fbxo4 target−cyclin D1 and hnRNPK revealed 10 aa fragment that overlaps with the D1 degron cassette. Identified fragment of high similarity and identity is flanked by two lysines. **F, G** Substitution of hnRNPK K21 and K34 reduced ubiquitylation significantly in vitro in 293T cells (**F**) and in vivo (**G**). **H** In vivo ubiquitylation assay in HEK293T cells after cytosolic/nuclear fractionation. SCF$^{Fbxo4}$-dependent hnRNPK polyubiquitylation is maintained in the cytoplasm.

hnRNPK levels (Fig. S4E, F). GO analysis also highlighted dysregulation in cytoskeleton rearrangement, often a hallmark of cellular motility (Fig. S4B, C). To address the role of SCF$^{Fbxo4}$-hnRNPK in the regulation of cell motility we seeded Fbxo4$^{+/+}$ or Fbxo4$^{-/-}$ MEFs at low density and tracked movements of ~150 individual cells by live microscopy for 10 h. Fbxo4$^{-/-}$ cells covered a significantly larger Euclidean distance relative to Fbxo4$^{+/+}$ (Fig. 4G). We also observed higher ratio of low velocity cells (red marks, threshold: <0.25 µm/min) for Fbxo4$^{+/+}$ cells. The motility of both groups was dramatically impaired by hnRNPK knock-down (Fig. 4G−I).

## SCF$^{Fbxo4}$-hnRNPK axis contributes to melanoma metastasis and progression

We next examined the relevance of the SCF$^{Fbox4}$-hnRNPK regulatory axis in melanoma. Fbxo4 inactivating mutations occur in ~10% of melanoma and a common mutation is I377M, inhibiting substrate binding[17]. Consistently, the I377M mutation attenuates Fbxo4-hnRNPK interaction (Fig. S5A) and diminished hnRNPK ubiquitylation (Fig. S5B). We further compared three melanoma cell lines bearing Fbxo4 mutations Lu1205$^{I377M-Fbxo4}$, WM739B$^{I377M-Fbxo4}$, WM3918$^{I377M-Fbxo4}$, two cell lines harboring FBXO4$^{wt}$ (WM983B$^{wt-Fbxo4}$, WM35$^{wt-Fbxo4}$) versus primary human melanocytes (PHM80 and PHM90). An increase in c-Myc protein was noted in Fbxo4 mutant cell lines relative to cells with wild type Fbxo4 (Fig. 5A). Critically, c-Myc protein was restored to wild type levels following hnRNPK knock-down (Fig. S5C). Consistent with observations in Fbxo4-deficient MEFs, Fbxo4-I377M melanoma cell lines exhibited increased capacity for invasion relative to melanoma cell lines with Fbxo4;$^{wt}$ this invasive phenotype was dependent on expression of hnRNPK (Fig. 5B). Subsequently, we knocked down Fbxo4 in WM983B to determine whether loss of Fbxo4 would reproduce a similar genomic landscape to that observed in Fbxo4$^{-/-}$ cells. Pathway analysis of RNA-seq revealed high representation of genes involved in ECM remodeling, focal adhesion, adherens junction, or integrin surface interaction (Fig. S5D−I).

Because the identified regulatory network of SCF$^{Fbxo4}$-hnRNPK highlights contributions to invasion and intravasation into local tumor vasculature rather than niches, we investigated early stages of metastasis by subcutaneous inoculation of GFP-expressing B16F10 melanoma cells in syngeneic C57BL/6 J mice. To address the SCF$^{Fbxo4}$-hnRNPK pathway, we either knocked-down hnRNPK or overexpressed Fbxo4/αB-crystallin which should inhibit hnRNPK-dependent increases in c-Myc translation. Control tumors achieved a significantly higher average volume relative to those with either hnRNPK knockdown or Fbxo4/αB-Crystallin overexpression (Fig. 5C). Because the larger tumor size might bias towards increased metastatic process assessed by GFP fluorescence in the lungs, we matched individual mice with similar sized primary tumor growth dynamics and end-point tumor size across the three conditions (Figs. 5D and S6A). A direct measurement of fluorescence from the whole lung shows lower signal in the lungs of mice bearing tumors with hnRNPK knock-down or Fbxo4/αB-crystallin overexpression (Figs. 5E, F and S6B). A similar result was observed using qPCR as an independent method to detect GFP in lung tissue (Fig. 5G). Histological analysis of tumors confirmed introduced genetic modification (Fig. S6C) and lung colonization (Fig. S6D). Collectively, these data suggest loss of hnRNPK or overexpress of Fbxo4 is sufficient to significantly suppress spontaneous lung colonization of B16F10 melanoma cells in vivo.

Our data demonstrate altered hnRNPK activity due to Fbxo4 loss results in increased c-Myc protein synthesis and c-Myc-dependent cell invasion (Fig. 4E, F). Considering that high c-Myc protein levels correlate with lower survival (Fig. S6E), we considered potential therapeutic strategies for exploiting c-Myc dependence. To deplete c-Myc we utilized JQ1, a common pharmacological tool for targeting c-Myc vulnerabilities. B16F10 melanoma cells were implanted subcutaneously and allowed to establish for 7 days; mice were separated into two groups for treatment with either JQ1 or vehicle for 14 days. We did not note a significant decrease in primary tumor volume in the JQ1 treated group (Figs. 5H and S6F). In contrast, lung colonization was significantly reduced with JQ1 (Fig. 5I−L). JQ1 did not reduce hnRNPK while it efficiently depleted c-Myc (Fig. 5M). Staining for cancer-specific proliferation marker Ki67 revealed a significant decrease in mitotic index (Fig. 5M). These data support a model wherein the SCF$^{Fbxo4}$-hnRNPK axis is a potent pro-metastatic signaling pathway and at least in part functions through c-Myc, highlighting the potential for using therapeutic strategies that impact c-Myc function.

## The SCF$^{Fbxo4}$-hnRNPK-c-Myc axis is altered in human cancers

In addition to mutations, Fbxo4 protein loss is observed at high frequency in melanoma and esophageal cancers[24,31]. This phenotype likely reflects dysregulation of Fxr1 and Fxr1-dependent inhibition of Fbxo4 synthesis in these cancers (Fig. 6A; ref. 21). Evaluation of Fxr1 expression data deposited in TCGA and GTEx databases highlights a significant increase of Fxr1 in melanoma patients (Fig. 6B). In contrast, no significant changes in hnRNPK expression were apparent emphasizing the importance of translational or posttranslational control over expression level (Fig. S6G). Likewise, αB-Crystallin is lost at high frequency in melanoma (ref 32. and Fig. 5A). Further evaluation of TCGA cancer genomic data collection reveals loss of copy number for the αB-Crystallin encoding gene *CRYAB* in over 15% of melanoma cases (Fig. 6C). To define whether our data can be interpreted in the clinical context, we performed IHC staining on melanoma patient tissue samples. Comparison of melanoma resection samples with normal human skin samples revealed increased c-Myc with reduced αB-Crystallin and Fbxo4 (Figs. 6D−G and S6H). We also noted a significant shift of hnRNPK to the cytoplasm, perhaps reflecting an indirect consequence of increased function in translating ribosomes observed in cancer cells (Fig. 6F).

We also evaluated esophageal squamous cell carcinoma (ESCC), a cancer where tumor development is associated with aberrant SCF$^{Fbxo4}$ signaling[31]. Oncomine cancer data provide evidence for Fxr1-Fbxo4 inhibitory engagement due to Fxr1 overexpression in ESCC at the mRNA level (Fig. 6H) and gain of copy number (Fig. 6I). Clinical correlation was addressed again by the immunohistochemical screen of ESCC cancer samples compared with adjacent normal esophagus. In ESCC we noted a remarkable cytoplasmic concentration of hnRNPK, reduced αB-Crystallin and Fbxo4 which correlates directly with elevated c-Myc (Fig. 6J−M). Collectively, these data support loss of SCF$^{Fbxo4}$ function allows for pro-tumorigenic activity of cytoplasmic hnRNPK which results in an increase of the c-Myc oncoprotein due to enhanced translation.

## Discussion

Under physiological conditions, cells maintain high levels of nuclear hnRNPK and low levels of cytoplasmic hnRNPK; both populations

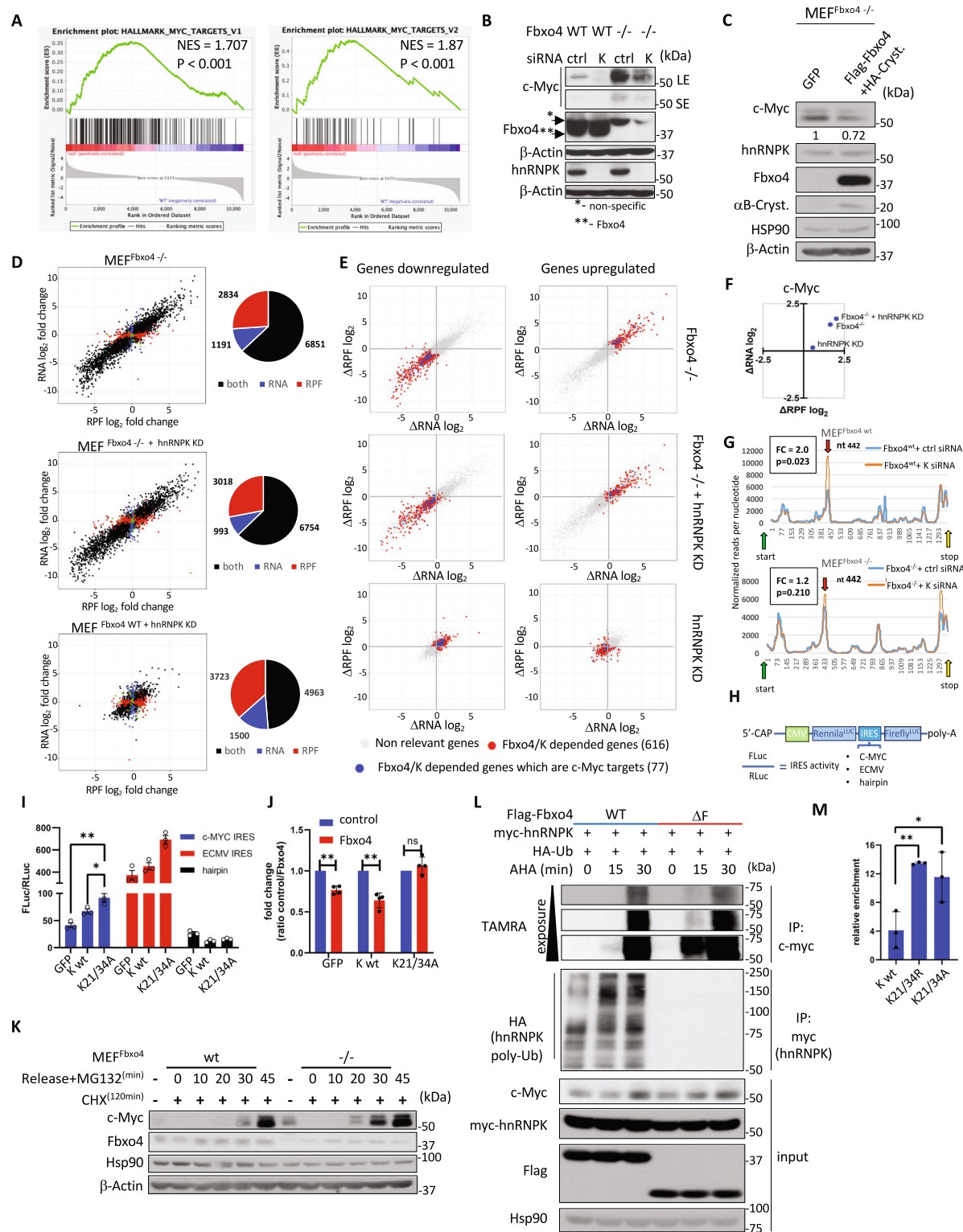

regulate mRNA turnover, translation or splicing, while the cytoplasmic population also engages in control of mRNA translation. Abnormal cytoplasmic accumulation of hnRNPK occurs frequently in cancer and is clinically unfavorable in various cancers. Whereas the cytoplasmic enrichment of hnRNPK via ERK- and CDK2-dependent phosphorylation[33,34] or loss of KI domain methylation[35] have been reported, the functional significance and mechanistic regulation of cytoplasmic hnRNPK and its contribution to tumor growth and progression remained poorly understood. Here we demonstrate that

pro-tumorigenic activities of cytoplasmic hnRNPK are checked by the tumor suppressive function of Fbxo4 (Fig. S7).

Our data demonstrate that SCF[Fbxo4] ubiquitylates cytoplasmic hnRNPK thereby limiting the capacity of hnRNPK to bind c-Myc mRNA and to support c-Myc translation. It is likely that additional hnRNPK target mRNAs, as highlighted in our ribo-seq, contribute to regulation of tumor cell growth and are critical to maintain cell homeostasis. Therefore, the inactivation or loss of Fbxo4 that occurs in melanoma and esophageal cancer unleashes cytoplasmic hnRNPK activity,

**Fig. 3 | SCF^Fbxo4–hnRNPK controls c-Myc synthesis and affects the genomic landscape. A** GSEA analysis of mRNA dysregulated in Fbxo4^-/- cells reveals c-Myc signature. **B** Fbxo4^-/- MEFs express a high level of c-Myc which is attenuated by hnRNPK depletion. **C** Re-introduction of Fbxo4 to Fbxo4^-/- MEFs rescues c-Myc protein levels. Signal intensity for c-Myc was adjusted to Hsp90. **D** MEFs were subjected to ribosome profiling and RNAseq in the following genetic conditions: (i) MEF^wt, (ii) MEF^Fbxo4-/-, (iii) MEF^wt + hnRNPK RNAi, (iv) MEF^Fbxo4-/- + hnRNPK RNAi. Conditions ii–iv were compared to (i) MEF^wt and summarized as scatter plot to express changes in translation efficiency, number of events occurred were presented in the pie charts. **E** The SCF^Fbxo4-hnRNPK dependent targets were selected based on the following criteria: (I) 2-fold up or down regulated in MEF;^Fbxo4-/- (II) rescued by >25% in MEF^Fbxo4-/- + hnRNPK knock-down; (III) regulated in MEF^wt + hnRNPK knock-down in the opposite direction to MEF^Fbxo4-/- or had no effect; blue dots represent c-Myc putative targets (detailed list is presented in Supplementary Data 1). **F** Regulation of c-Myc in different SCF^Fbxo4/hnRNPK setting compared to MEF^wt. **G** Comparison of RPF readouts across c-Myc transcript quantified as a normal reads per nucleotide; arrows indicate translation start (green) and stop sites (yellow). **H–J** Bi-cistronic luciferase reporter system was used to evaluate c-Myc IRES activity (ECMV IRES- luciferase signal positive control; hairpin - negative control) in NIH3T3 cells. **K** CHX-driven depletion followed by release with proteasome inhibitor to compare c-Myc synthesis rate in MEF^Fbxo4-/- versus MEF^wt. **L** Subsequent evaluation of hnRNPK ubiquitylation and c-Myc synthesis rate was performed in HEK293T cells. **M** RNA immunoprecipitation assay followed by qPCR (MEF cells). The data in **I, M** represents mean ± SD and was analyzed by two-tailed Student's $t$ test ($n = 3$). The data in **J** represents mean ratio (control/Fbxo4) and was compared by ratio paired Student's $t$ test ($n = 4$). In **E**, all selected targets MEF^Fbxo4-/- versus MEF^wt complied with adj. $p$ value <0.05 for either RNA and RPF, the rescue comparisons were included based on the fold change. Data presented in the **A, D–G** is based on the RNA-seq and Ribo-seq data run in biological duplicates. Exact $p$ values from left to right, **I** $p = 0.0084$, $p = 0.0320$; **J** $p = 0.0036$, $p = 0.0096$, $p = 0.3182$; **M** $p = 0.0034$, $p = 0.0419$.

unmasking malignant function though dysregulated translation of multiple mRNA targets that promote tumor progression. While many retrospective studies focus on the modest increase in total hnRNPK that occurs in cancer, this modest increase is likely secondary to the role of dysregulated cytoplasmic hnRNPK. HnRNPK is embryonically essential[36] and constitutively accumulates at high levels in the nucleus reflecting its low rate of protein degradation ($t_{1/2} > 10$ h). Not surprisingly, mutations in hnRNPK are rare. Nucleocytoplasmic redistribution and PTM-dependent regulation are logically the primary determinants of hnRNPK pro-malignant activity. These features are consistent with the general genomic landscape of RBPs in cancer that reveal rare RBP mutations and relatively small changes in RBP mRNA levels[37–39]. Indeed, collectively, our data provide additional view of the tumorigenic properties of hnRNPK. While cytoplasmic accumulation has been correlated with cancer, no mechanistic insights regarding how cytoplasmic hnRNPK is regulated or how it functions in a pro-tumorigenic pathway have been provided. Our results demonstrate that cytoplasmic hnRNPK contributes to recruitment and efficient translation of key mRNAs such as c-Myc. Ubiquitylation of hnRNPK by SCF^Fbxo4 is a critical modification that limits hnRNPK's ability to drive protein output. Loss of SCF^Fbxo4 results in unrestricted hnRNPK activity, overexpression pro-oncogenic factors (eg. c-Myc), increased proliferation, invasion and motility leading to an aggressive cancer phenotype. While c-Myc is not the only target, it represents a key oncogenic driver and potential therapeutic target in cancers with loss of Fbxo4 and accumulation of cytoplasmic hnRNPK.

Mechanistically, SCF^Fbxo4-dependent polyubiquitylation of hnRNPK through K63 linkages is distinct from other known SCF^Fbxo4 substrates which use K48 linkages to target degradation. We note that SCF^Fbxo4 preferentially utilizes UbcH5, an E2 that utilizes both K48 and K63 linkages (Fig. S2K; ref. 40). It is striking that ubiquitylation of lysines within hnRNPK did not detectably interfere with nuclear import. This likely reflects the importance of the second nuclear import sequence in hnRNPK. Critically, ubiquitylation of KHSRP and p53 within NLS, had non-proteolytic outcomes and did not impact nucleocytoplasmic distribution[41,42].

HnRNPK loss triggers reduced c-Myc protein levels; paradoxically this corresponds with a simultaneous increase of ribosome coverage to specific regions of c-Myc mRNA. The divergence between protein accumulation and ribosome coverage at specific sequences is often characteristic of ribosome pausing. Our data support a model wherein non-ubiquitylated hnRNPK binds c-Myc mRNA and facilitates translation initiation at the 5'UTR. During elongation, the continued presence of hnRNPK is needed to prevent ribosome pausing. Indeed, the increased peaks for ribosomes on c-Myc mRNA (Fig. 3G) in the absence of hnRNPK reflects ribosome pausing. Ubiquitylation of hnRNPK reduces its binding to target mRNA, such as c-Myc, and reduces translation initiation which reflects the low RNA binding by Ub-hnRNPK. Ubiquitylation of hnRNPK limits its availability for bypassing ribosome pausing on transcripts that need hnRNPK for translation elongation. Importantly, c-Myc possess two independent regions previously discovered as pausing sites[43,44]. Translation through pausing sites requires recruitment of specific factors like eIF5A[43]. It is tempting to speculate that in addition to regulation of RNA binding, ubiquitylated hnRNPK may regulate association with translation factors such as eIF5 to regulate translation efficiency. Our data support a model wherein SCF^Fbxo4–hnRNPK controls c-Myc protein levels through regulation of protein synthesis. Ubiquitylation of hnRNPK inhibits its general RNA-binding ability. For a specific subset of targets, eg. c-Myc, hnRNPK ubiquitylation will result in reduced polysomal recruitment and reduced c-Myc mRNA translation. In contrast, in cancer cells lacking Fbxo4 or/and αB-crystallin, unmodified hnRNPK recruits c-Myc mRNA and supports its efficient translation.

Genomic analysis highlights the role of the SCF^Fbxo4–hnRNPK axis for cell motility and invasiveness. To elucidate the mechanistic underpinnings, we assumed hnRNPK exhibits a multidimensional activity and its pro-oncogenic function is suppressed by SCF^Fbxo4. In terms of compartmentation, identified targets were associated with extracellular or outer membrane activity and functionally implicated in the regulation of matrix, intracellular interaction, and cytoskeleton rearrangements. The genomic landscape revealed pathways that contribute to matrix remodeling, invasion and intravasation all key aspects of tumor metastasis. Consistently, our mechanistic studies demonstrate that loss of Fbxo4 increases cell motility, cell invasion, and melanoma metastasis in an hnRNPK- and c-Myc-dependent manner. These results support case-control reports revealing a correlation between advanced tumor staging and hnRNPK cytoplasmic localization, and provide a mechanism that defines the robust, Fbxo4-dose dependent, metastatic spread in Fbxo4 knock-out mice that express the BrafV600E oncogene in melanocytes[17].

Cancers of the esophagus also exhibit hnRNPK mislocalization and SCF^Fbxo4/αB-Crystallin loss. Analysis of primary ESCC revealed remarkable accumulation of cytoplasmic hnRNPK relative to normal adjacent tissue; this corresponded with increased c-Myc and concomitant reduction in Fbxo4 and its cofactor αB-Crystallin. These results emphasize the broad significance of SCF^Fbxo4-hnRNPK regulation for tissue homeostasis and imply potential value of Fbxo4/αB-Crystallin/hnRNPK/c-Myc axis status for prognosis.

Among hundreds of targets that emerged in bioinformatic analyses, c-Myc was of particular interest not only as a prominent oncoprotein, but due to significant recent advances in the development of c-Myc-targeted therapeutic strategies. One such strategy, currently being evaluated in clinical trials, exploits inhibition of c-Myc transcription activators−Bromodomain extra-terminal (BET) proteins (BRD2, 3, 4); BRD4 inhibitors clinical impact is thought to reflect inhibition of c-Myc expression and thus c-Myc function[30]. Our work demonstrates that

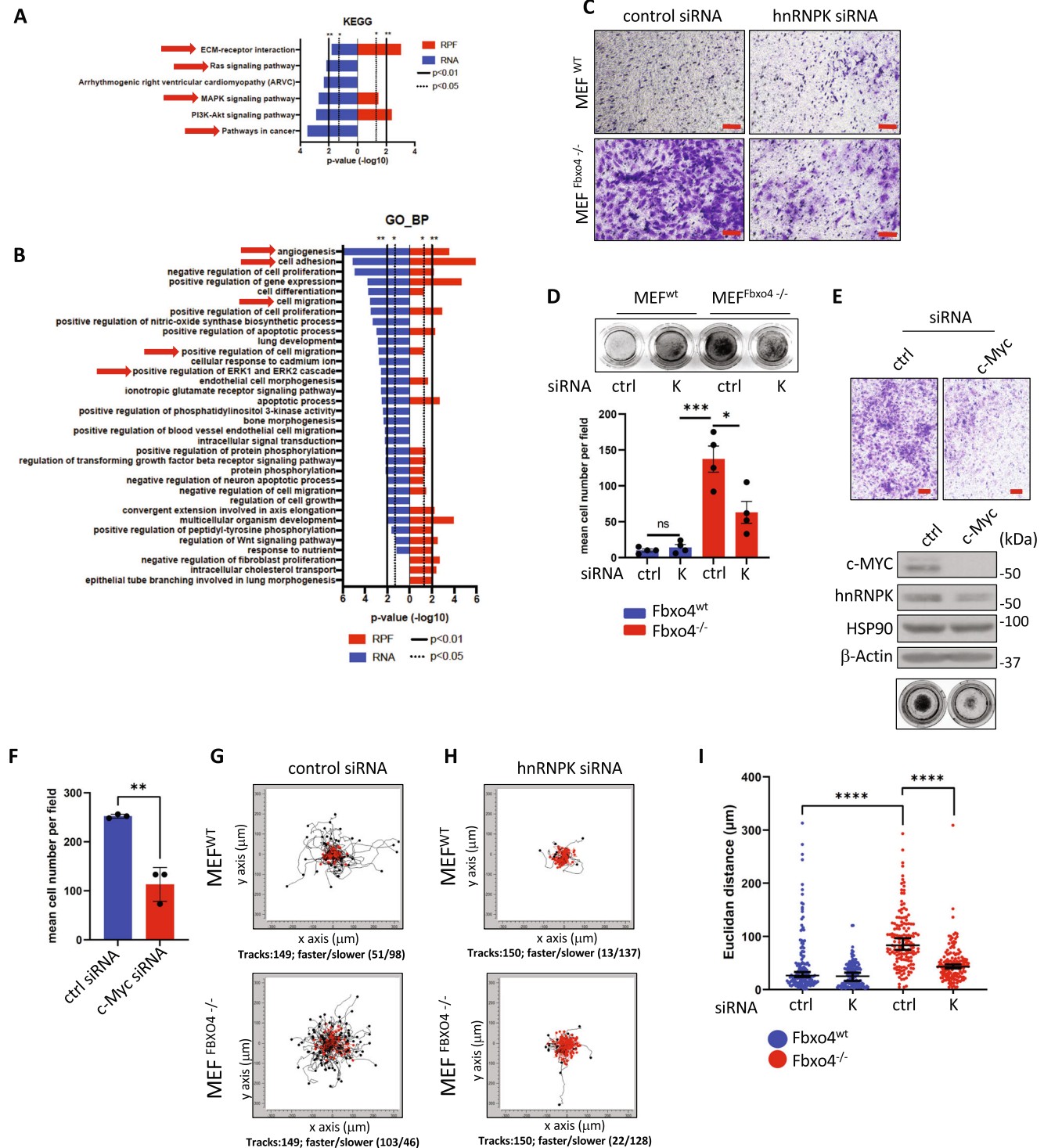

**Fig. 4 | SCF^Fbxo4 – hnRNPK – c-Myc axis regulates cell invasion and motility.**
**A, B** The population of targets identified in Fig. 3E was subjected to functional analysis by KEGG and Gene Ontology- biological process annotation (GO_BP). Data was evaluated by Fisher's exact test $p < 0.01$ are considered significant.
**C** Representative images of cell invasion through extracellular matrix in Fbxo4 deficient cells with or without hnRNPK RNAi depletion; **D** Boyden chamber images and quantification of invasion assay from **C. E** Representative images from invasion assay and western blot showing the effect of c-Myc siRNA-mediated knock down on MEF^Fbxo4−/−^ cells invasiveness. **F** Quantification of **E** expressed as a mean cell number.

**G, H** present tracks of individual cells in cell tracking assay. Red paths indicate on cells moving <0.25 μm/min **I** Quantification of motility in covered Euclidean distance. The data from invasion experiments (**D, F**) represents mean ± SD and was analyzed by two-tailed Student's *t* test (**D**−*n* = 4, **F**−*n* = 3). In cell motility assay, 149/150 cells were tracked per condition in three biologically independent experiments (49 or 50 cells/experiment). Data represents median with 95% CI and were compared by two-tailed Student's *t* test. *$p < 0.05$, **$p < 0.01$, ***$p < 0.001$, ****$p < 0.0001$. Scale bars **C, E**−100 μm. Exact *p* values from up to down: **D** $p = 0.0006$, $p = 0.026$, $p = 0.3892$; **F** $p = 0.0023$; (**I**) $p = <0.0001$ (both).

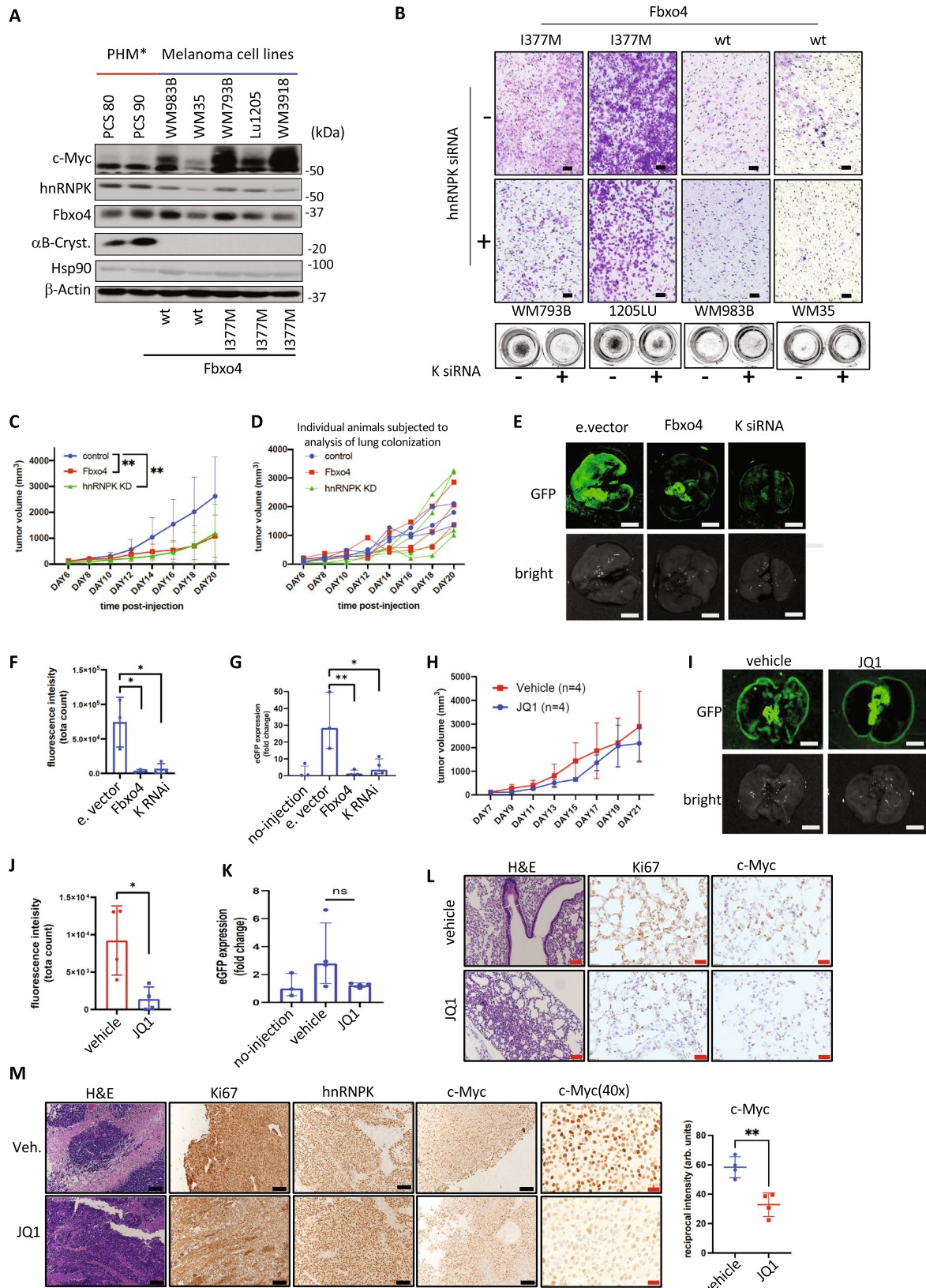

administration of JQ1 (BRD4 inhibitor) successfully depleted c-Myc levels with a striking impact on tumor spread/metastasis with only minor inhibitory activity on the primary tumor. We specifically noted decreased lung metastatic colonization suggesting potential therapeutic vulnerability toward prevention of melanoma metastasis. This is consistent with previous reports highlighting little to no effect of JQ1 on

volume or weight of primary melanoma tumors[45]. JQ1 also decreased c-Myc in multiple ESCC cell lines having significant impact on their invasion and ESCC xenograft tumor growth[46]. Such results should be viewed with caution, however, as certain BRD4 activity is not limited to c-Myc. Therefore, the observed inhibition of metastasis likely reflects cooperative depletion of several BRD4 targets.

**Fig. 5 | SCF^Fbxo4^-hnRNPK axis contributes to melanoma progression. A** c-Myc protein in multiple melanoma cell lines in comparison to normal primary human melanocytes. **B** Representative pictures of trans-well invasion assay on melanoma cell lines in the context of Fbxo4 (I377M mutant vs wt) and hnRNPK (hnRNPK vs control siRNA) status. **C** The average tumor volume increase across the time is in melanoma allograft model with subcutaneously injected B16F10 cells modified by (i) overexpression of Fbxo4/aB-Crystallin, (ii) hnRNPK knock-down or (iii) control cells. **D** Dynamics of tumor volume increase in mice chosen for lung colonization assessment. **E** Lung colonization was measured by detection of GFP signal with background normalization to non-injected mice, **F** fluorescence intensity was quantified by average signal from 3 non-overlapped spots. **G** Expression of GFP was confirmed by qPCR performed on total mRNA lung extract from lung tissue compared to mRNA isolated from non-injected mice. **H** Tumor volume increase in B16F10 melanoma allograft model upon transcriptional depletion of c-Myc by administration of BRD2/4 inhibitor, JQ1. **I, J** show representative pictures of GFP

signal presented in and its quantification analogically to E, **F. K** Quantification of eGFP mRNA level in JQ1 treatment as an indicator of lung colonization compared to mice treated with vehicle. **L** Representative pictures of Ki67 and c-Myc IHC staining on lung from JQ1 treatment experiment. **M** Histological analysis of primary tumor from JQ1 treatment experiment; graph represent quantification of DUB signal from c-Myc staining. All data represents mean ±SD and was analyzed by Two-way ANOVA with Geisser–Greenhouse correction (**C, H**; $n = 10$ and 4, respectively) or two-way Student's $t$ test (**F, G, I, K**), *$p < 0.05$, **$p < 0.01$. Analysis of **F** and **J** was done on $n = 3$ for empty vector, $n = 3$ for Fbxo4 OE, and $n = 4$ for K RNAi. Analysis of **J** and **K** was run on $n = 4$ for vehicle and JQ1 treated group. C-Myc quantification (**M**) was done on 3 representative picture ×40 from each staining using ImageJ and reciprocal intensity approach (maximum intensity−measured intensity; arb.units.−arbitrary unit. Scale bars: black−100 μm, red−20 μm. Exact $p$ values from up to down: **C** $p = 0.0098$, $p = 0.0071$; **F** $p = 0.0229$, $p = 0.0275$; **G** $p = 0.0270$, $p = 0.0088$; **H** $p = 0.3519$; **J** $p = 0.0365$; **K** $p = 0.0552$; **M** $p = 0.0033$.

Given genetic depletion of hnRNPK effectively inhibited metastasis, therapeutic targeting of hnRNPK could also be considered. To date no direct chemical inhibition strategy for hnRNPK has been developed. Similarly to c-Myc, hnRNPK can be indirectly downregulated at the level of transcription by Mithramycin, which targets the SP1 transcription factor[47]. Indeed, Mithramycin was subjected to phase I/II clinical trial in Edwin's Sarcoma; however, trials were terminated early due to associated toxicity[48].

Our studies reveal a dual nature of the SCF^Fbxo4^ E3 ubiquitin ligase complex wherein it can either regulate protein turnover through K48 ubiquitin linkages (Pin1/Trf1, cyclin D1, Fxr1) or specifically modify substrate function (hnRNPK) through K63 linked polyubiquitylation. It is noteworthy, that all established relationships between SCF^Fbxo4^ and its substrates, phenotypically reflect a tumor suppressor (Fbxo4)−oncogene (substrate) interplay. This creates a consistent picture of SCF^Fbxo4^ as critical regulator of tumor development affecting uncontrolled proliferation, senescence bypass (D1, Fxr1) and tumor progression (hnRNPK).

Beyond hnRNPK, Fbxo4 co-purified with a multiple KH-domain containing RBPs. We note that transcripts differentially regulated upon loss of Fbxo4 are enriched in binding motifs for many of the RBPs that co-purify with Fbxo4 suggesting a much broader relationship between SCF^Fbxo4^, RBP and the regulation of RNA metabolism. Elucidation of the regulatory relationships between Fbxo4 and RBPs will provide key insights into the regulation of cell homeostasis by Fbxo4, its role as a tumor suppressor and hopefully contribute to the identification of new therapeutic opportunities for cancer treatments.

## Methods

This study has been conducted in accordance with international and local ethical standards. Murine model research was approved by IACUC (2019-0052) and maintained in the Animal Resource Center at Case Western Reserve School of Medicine following all relevant ethical guidelines.

### Cell culture

HEK293T cells were purchased from ATCC in 2014 and cultured in DMEM supplemented with 10% FBS (Gemini) and Penicillin-Streptomycin (Corning). Human osteosarcoma U2OS cells (ATCC HTB-96) were purchased in 2014 from American Type Culture Collection (ATCC) and maintained in McCoy's 5a medium (Corning) supplemented with 10% of FBS and Penicillin-Streptomycin. Lung carcinoma epithelial cells A549 were purchased in 2019 from American Type Culture Collection (ATCC) and maintained in F-12K Medium (ATCC Cat# ATCC 30-2004) supplemented with 10% FBS and Penicillin-Streptomycin. TE15 ESCC cells were kindly provided by Dr. Tetsuro Nishihara who established this cell line and all cells were maintained in RPMI1640 with 10% FBS and 1% penicillin-streptomycin. Authenticated human melanoma cell lines (1205Lu, WM983B, 451Lu, WM3918, and

WM35) were obtained from the Wistar Institute collection (Philadelphia, PA) and cultured in media containing: 80% MCDB153 medium, 20% Leibovitz L-15, 2% fetal bovine serum (FBS), 1.68 mM $CaCl_2$ and Penicillin-Streptomycin. Primary human melanocytes derived from two independent individuals were purchased from ATCC (cat#PCS-200-013) and cultured in Dermal Cell Basal Medium (ATCC cat#PCS-200-030) supplemented with Adult Melanocyte Growth Kit components (ATCC cat#PCS-200-042). B16F10 melanoma cell line was cultured in DMEM supplemented with 10% FBS and Penicillin-Streptomycin. Sf9 cells were purchased from Gibco in 2018 and maintained as suspension culture in Grace's media (Gibco cat#B82501) supplemented with 10% FBS. NIH3T3 cells were purchased from ATCC in 2019 from ATCC and cultured in DMEM supplemented with 10% FBS and Penicillin−Streptomycin. MEF cells were derived from Fbxo4^+/+^ or Fbxo4^−/−^ transgenic mice embryo (B57BL/6 background developed by Vega Biolab, PA) at day 14 of gestation applying 3T9 passaging protocol in the DMEM media supplemented with 10% FBS, 55 μM β-mercaptoethanol, 2 mM glutamine, 0.1 mM nonessential amino acids and 10 μg/ml gentamicin. All cell lines were confirmed to be mycoplasma free using Universal Mycoplasma Detection Kit (ATCC, cat# 30-1012K).

### Plasmids and viral vectors

Flag-Fbxo4 and its mutants were encoded in pCDNA3.1 backbone vector. Untagged pCMV6-AC hnRNPK plasmid was purchased from Origene (cat# SC321563) and modified to (1) insert N-terminal myc peptide sequence, (2) delete KI domain, (3) delete N-terminal region, (4) point mutation at K21 or/and K34 with the use TaqMaster Site-Directed Mutagenesis Kit (GM Bioscience, cat#7001). A retroviral system was used to overexpress hnRNPK, Fbxo4 or αB-Crystallin in MEFs or melanoma cell lines. Virus was produced in HEK293T cells by co-transfection of helper plasmid ψ2 with either pmx-Flag-Fbxo4, pmx-αB-Crystallin or hnRNPK ORF cloned into MigR1 vector. Anti-Fbxo4 shRNA constructs were encoded in the pLKO1 plasmid that was utilized to produced lentivirus particles in HEK293T cells by liposomal co-transfection along with the following helper plasmids: pMDL, pRSV and pVSV-G. For infection, lentivirus or retrovirus containing media was incubated with cells for 8–16 h in the presence of 10 μg/ml Polybrene (Millipore cat# TR-1003-G, Germany). Baculovirus was generated with the use of Bac-to-Bac system (ThermoScientific) according to the manufacturer's guidelines; briefly, hnRNPK ORF was cloned into pFastBac1 vector, an expression cassette was next transposed into baculoviral shuttle vector (bacmid) by transformation of MAX Efficiency DH10Bac E. coli. To induce virus production recombinant bacmid was isolated and delivered to the Sf9 insect cells by transfection with TransIT reagent (Mirus, cat#Mir6104), 7 days post-transfection virus was collected and applied in further procedures. pEF-HA-ubiquitin plasmids encoding wt, K48R or K63R ubiquitin variants were used to perform in vivo ubiquitylation assays. pTRIP bi-cistronic luciferase reporter vectors encoding EMCV IRES, c-Myc IRES or hairpin

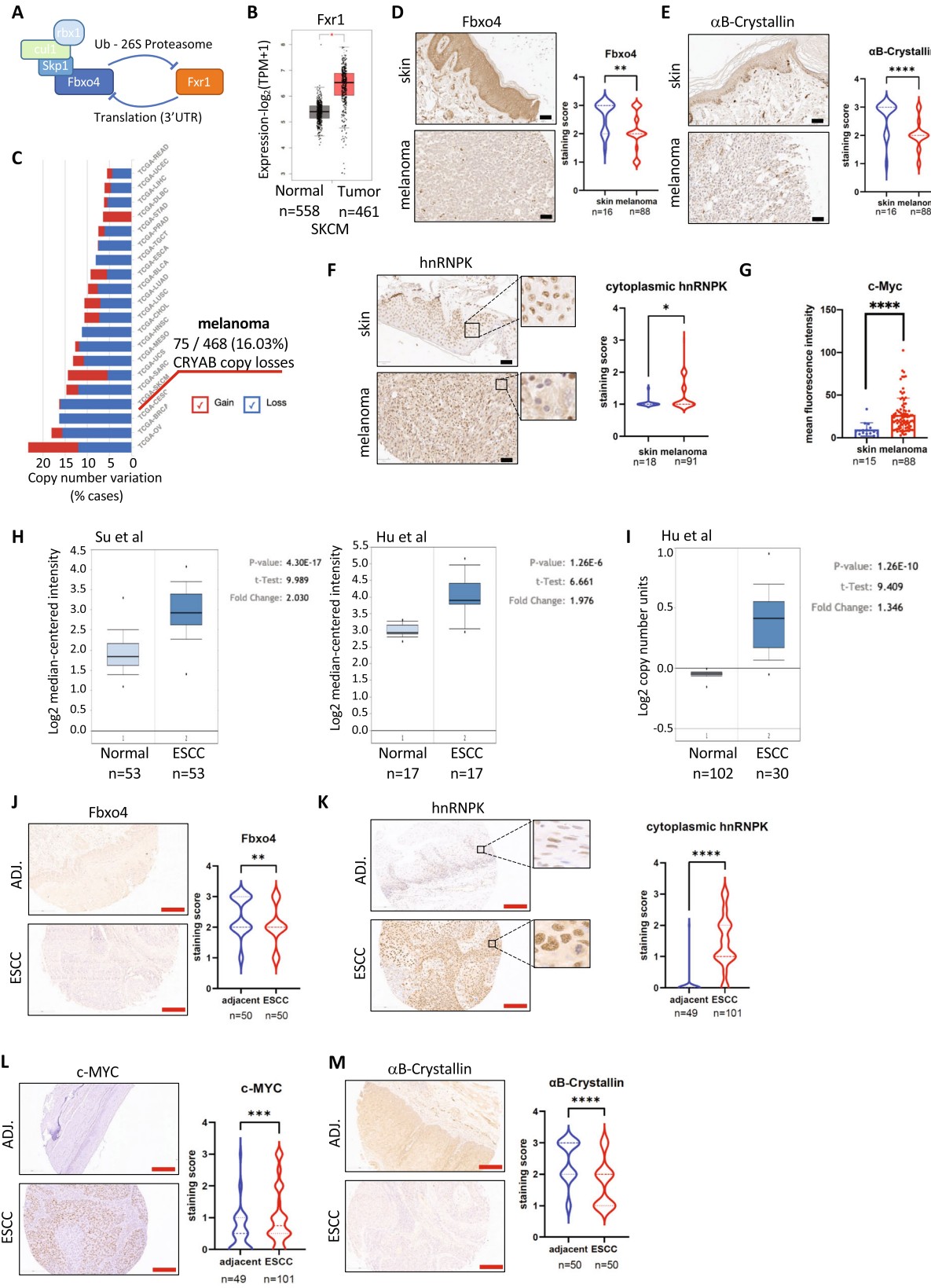

were a kind gift from Dr. Anne-Catherine Prats (Institut des Maladies Métaboliques et Cardiovasculaires, France).

**Human tissue microarrays**

Melanoma tissue microarrays containing normal skin samples as reference- ME2081 and T386a were purchased from US Biomax, Inc.

and utilized to detect the following proteins, ME2081- Fbxo4, αB-Crystallin, c-MYC, hnRNPK; T386A- hnRNPK. Esophageal squamous cell carcinoma (ESCC) tissue microarrays described previously[49] were stained for c-Myc and hnRNPK. ESCC tissue microarrays purchased from US Biomax Inc. (ES1505) were stained for Fbxo4 and αB-Crystallin. Immunofluorescence was used to stain c-Myc on the

**Fig. 6 | SCF^Fbxo4-hnRNPK regulation is impaired in human cancers. A** Fxr1 and Fbxo4 are mutual negative regulators. **B** GTEx and TCGA data shows increased Fxr1 expression in melanoma. CRYAB expression in TCGA SKCM ($n = 558$) compared to the normal ($n = 461$, TCGA and GTEX) generated in the GEPIA2 tool. Both groups were compared by one-way ANOVA. Boxplot center represents median, bounds represent 25 and 75%, and whiskers show the minimum or maximum no further than 1.5 times interquartile range from the bound. **C** Gene copy variation of *CRYAB* (αB-Crystallin protein coding) from TCGA database analyzed by NIH Genomic Data Commons Portal v1.28.0 (https://portal.gdc.cancer.gov/). **D–F** Immunohistochemical staining of melanoma and normal skin TMA with anti-Fbxo4, αB-Crystallin, hnRNPK antibodies. **G** Immunofluorescence staining of melanoma and normal skin TMA presented as a mean fluorescence intensity ±SD from 200 randomly selected cells/individual and statistically

compared by two-tailed Student's *t* test (normal/cancer $n = 15/88$). **H**, **I** Oncomine data base shows expression and copy number variation of Fxr1 in esophageal squamous cell carcinoma (reported by Su et al. and Hu et al.); **J–M** Immunohistochemical staining of ESCC and adjacent tissue TMA with anti-Fbxo4, and αB-Crystallin, c-Myc, hnRNPK antibodies. In IHC quantification of all TMAs the 0–3 scale was applied (0–negative staining, 1–low, 2–med, 3–high); results were summarized in violin plots presenting distribution of score and analyzed statistically by non-parametric two-tailed Mann–Whitney test (control/cancer; **D**, **E**–$n = 18/91$; **F**–$n = 18/91$; **J**, **M**–$n = 50/50$; **K**, **L**–$n = 50/50$). *$p < 0.05$, **$p < 0.01$, ***$p < 0.001$, ****$p < 0.0001$. Scale bars: black 50 µm; red 200 µm. Exact *p* values: **D** $p = 0.0023$; **E** $p = <0.0001$; **F** $p = 0.0296$; **G** $p = <0.0001$; **J** $p = 0.0092$; **K** $p = <0.0001$; **L** $p = 0.0006$; **M** $p = <0.0001$.

melanoma tissue microarray (ME2081). All remaining proteins were analyzed by immunohistochemistry and scored by certified clinical pathologist using 0–3 scale (0–negative, 1–low, 2–med, 3–high). All used samples were de-identified.

### Primers
**qPCR primers.** c-MYC: FRW-5′-CCCTATTTCATCTGCGACGAG-3′ REV-5′-GAGAAGGACGTAGCGACCG-3′; PTGER2: FRW-5′-GCCAGGAGAATGA GGTGGTC-3′ REV-5′-CCTGCTGCTTATCGTGGCTG-3′; eGFP: FRW-5′-TC CTTGAAGTCGATGCCCTT-3′ REV-5′-CACATGAAGCAGCACGACTT-3′ GAPDH: FRW-5′-AACAGCAACTCCCACTCTTC-3′ REV-5′-CCTGTTGCTG TAGCCGTATT-3′.

**Mutagenesis primers.** hnRNPK_K21R: FRW-5′-TGAATTTGGTAGACGCC CTGCAGAAGAT-3′ REV-5′-ATCTTCTGCAGGGCGTCTACCAAATTCA-3′; hnRNPK_K34R: FRW-5′-GGAACAAGCATTTAGAAGATCTAGAAACAC-3′ REV-5′-GTGTTTCTAGATCTTCTAAATGCTTGTTCC-3′; hnRNPK_K21/ 34R: FRW-5′-GGTAGACGCCCTGCAGAAGATATGGAAGAGGAACAAGCA TTTAGAAGA-3′ REV-Reverse 5′-TCTTCTAAATGCTTGTTCCTCTTCCAT ATCTTCTGCAGGGCGTCTACC-3′; hnRNPK_K21/34A: FRW-5′-GGTGCA CGCCCTGCAGAAGATATGGAAGAGGAACAAGCATTTGCAAGA-3′ REV-5′-TCTTGCAAATGCTTGTTCCTCTTCCATATCTTCTGCAGGGCGTGCA CC-3′; hnRNPK_del_NLS: FRW-5′-AACCAATGGTGAATTTGGTAACACTG ATGAGATGG-3′ REV-5′-CCATCTCATCAGTGTTACCAAATTCACCATTGG TT-3′; hnRNPK_del_KI: FRW-5′-ATGATTATGGTGGTTTTAGAAGAGGGA GACCT-3′ REV-5′-AGGTCTCCCTCTTCTAAAACCACCATAATCAT-3′.

### RNAi knock down
Knock-down of hnRNPK or c-Myc was induced by ON-target siRNA purchased from Dharmacon-Horizon (hnRNPK mouse cat#L-048002-01-0020; hnRNPK human cat#L-011692-00-0010; c-Myc mouse cat#L-0040813-00-0010). siRNA was delivered to the cells using Lipofectamine RNAiMAX transfection reagent (ThermoFisher, cat#13778075) according to the manufacturer's guidelines. Control samples were simultaneously transfected with non-targeting siRNA (Dhamacon-Horizon, cat#D-001810-01-20). Fbxo4 knock-down was performed by shRNA described previously[21].

### Animal model
All animals were housed in the Animal Resource Center in Case Western Reserve School of Medicine, mice were maintained in the ventilated micro-isolation cage (maximum 5 mice per cage) with the following conditions: temperature 65–75°F; 40–60% humidity; 12 h light/dark cycle with non-disturb dark cycle; constant access to water and food. For the experiment presented in Fig. 5C–G, 30 6-week-old C57BL/6J mice were purchased from The Jackson Laboratory (15 males, 15 females). B16F10 cells were infected with eGFP expressing retroviral vector. Knock-down of hnRNPK was induced by RNAi transfection, while Fbxo4/αB-Crystallin overexpression was induced by retroviral infection. For subcutaneous injections, $2.5 \times 10^5$ cells were suspended in a 1:1 mixture of DPBS and Matrigel Matrix

(Corning, cat# 354234). Beginning at 7 days post-injection, tumor volume was monitored by electronic caliper measurement every other day. At day 20, all mice were sacrificed to extract lung and primary tumor for further analysis. For JQ1 treatment, $2.5 \times 10^5$ GFP-labeled melanoma B16F10 cells were transplanted subcutaneously to six-week-old C57BL/6J mice (4 males, 4 females). 7 days after inoculation, the cohort was divided into two groups for 14 days of treatment with JQ1 inhibitor (MCE, HY-13030) or vehicle (5% DMSO in 95% (20% SBE-β-CD in saline)). Drug was administered daily by intraperitoneal injection with a dosage of 50 mg/kg of body weight. Tumor size was monitored simultaneously by caliper-based tumor volume measurement every other day. 21 days post-injection mice were sacrificed to extract the lung for evaluation of lung colonization. Lung colonization was assessed by GFP fluorescence macro-imaging of whole organ using CRIs Maestro imaging system (CRI Medical Devices) with the signal gain time of 2 s and background adjustment to lung from mice without cancer cell implantation. Signal was quantified as mean of total fluorescent count measured on 3 representative areas on the surface of each lung lobe. Fragments of each analyzed lung were homogenized (60 s with TissueRuptor II on ice) in QIAzol Lysis Reagent to extract RNA which was subsequently used to quantify spontaneous metastasis by qPCR (RNA purification, cDNA synthesis as described in the qPCR section) according to modified method described previously[50]; here PCR reactions were performed with the primers specific to GFP. All procedures were performed in accordance with protocol approved by IACUC (2019-0052) at the Case Western Reserve University (CWRU). The maximum tumor volume allowed under the protocol ($1 \text{ cm}^3$) had been exceeded during the experiment. We determined that a 3-week primary tumor growth, although it led to excessive tumor size, allowed for parallel monitoring of tumor volume size and lung colonization. We found this strategy to be less burdensome by virtue of limiting the number of mice (two parameters measured in one experiment) and avoiding the more involved procedure of surgical removal of primary tumor followed by metastasis foci growth over the next 9–12 weeks. All mice were monitored daily and showed no severe impairment in physiological and neurological function.

### RNA Immunoprecipitation
MEF cells were infected with a retroviral vector to induce expression of GFP, myc-hnRNPK wt, myc-hnRNPK K12/34R or K21/34A. 72 h post-infection -20 mln cells were collected to perform precipitation of myc-hnRNPK–RNA complexes with the use of Magna RIP Kit (Sigma Aldrich cat#17-200). The quality of isolated RNA was evaluated by Bioanalyzer (Agilent) and only samples with high integrity (RNA Quality Number, RQN > 7) were used in the further procedures. Generated by iScript cDNA Synthesis Kit (Bio-Rad), cDNA was analyzed by real-time PCR using c-Myc specific primers. The enrichment of c-Myc in RIP samples was adjusted to c-Myc expression measured in the input RNA sample isolated from 10% of initial lysate and normalized to the sample expressing ectopic GFP.

## CHX chase assay

Cells were plated 24 h prior to cycloheximide (CHX) treatment to reach ~80% confluency. Cells were treated with 100 µg/ml of CHX (Sigma-Aldrich) and collected at indicated time points. Cell pellets were lysed in EBC buffer (50 mM Tris-Cl pH = 8, 120 mM NaCl, 0.5% IGEPA-CA630) and subjected to immunoblot analysis.

## $^{35}$S pulse labeling

NIH3T3 cells were either infected with myc-hnRNPK wt or ubiquitin mutant followed by transfection with non-targeting or hnRNPK specific siRNA. Forty-eight hours after infection/transfection procedure, cells were plated on 24-well plates at $1.2 \times 10^5$ cells/well, grown for additional 16 h, and incubated with [35S]Met/Cys (30 mCi/ml EXPRE35S Protein Labeling Mix, PerkinElmer) for 30 min. Cells were rinsed twice with cold phosphate-buffer saline (PBS) and total proteins were precipitated three times in 5% trichloroacetic acid (TCA) with 1 mM Met for 10 min on ice each time. Precipitates were dissolved in 200 µl of 1 N NaOH and 0.5% sodium deoxycholate for 1 h. Radioactivity was determined by liquid scintillation counting. Total cellular proteins were quantified with DC Protein Assay (Bio-Rad) following the manufacturer's instruction. Translation rates were normalized to the total protein content and expressed as an incorporation of [$^{35}$S] Met/Cys.

## qPCR

RNA was extracted by Qiazol Lysis Reagent (Qiagen cat#79306) according to manufacturer's manual. cDNA was obtained by reverse transcription performed with iScript cDNA Synthesis Kit (Bio-Rad, cat# 1708891). All quantitative-PCR reactions were run with SsoAdvanced Universal SYBR Green Supermix (Bio-Rad, cat# 1725270) with 500 nM concentration of primers and 10–100 ng of cDNA.

## In vitro ubiquitylation assay

SCF$^{Fbxo4}$ complex was isolated form Sf9 cells according to the procedure described previously[21]. Myc-hnRNPK was expressed by baculoviral infection of Sf9 cells which were subsequently lysed in Tween 20 buffer (50 mM HEPES, pH 8.0, 150 mM NaCl, 2.5 mM EGTA, 1 mM EDTA and 0.1% Tween 20) for purification of hnRNPK using anti-myc Agarose Affinity Gel affinity gel (Sigma-Aldrich, cat#A7470) followed by competitive myc peptide elution (Sigma-Aldrich, cat#M2435). Myc-hnRNPK and SCF$^{Fbxo4}$ were incubated with 50 ng of E1 ligase (Enzo, cat#BML-UW9410-0050), 500 ng of UbcH5c-E2 ligase (Enzo, cat#BML-UW9070-0100), 5 mM Mg-ATP (Enzo, cat# BML-EW9805-0100), 1 unit of inorganic pyrophosphatase (Sigma-Aldrich), 25 µg of ubiquitin, 5 µg of methylated-ubiquitin (Enzo, cat#BML-UW8555), and 1 µg of ubiquitin-aldehyde (Enzo, cat#BML-UW8450-0050) for 1 h at 37 °C. The reaction was stopped by addition of SDS-PAGE buffer and heating for 10 min at 95 °C. Samples were next subjected to immunoblot analysis.

## In vivo ubiquitylation assay

Ectopic expression myc-hnRNPK, Flag-Fbxo4, αB-Crystallin and HA or His tagged ubiquitin was induced in HEK293T cells by transient transfection using PolyJet reagent (SignaGen). 36 h post-transfection cells were lysed in NP40 buffer (50 mM Tris pH = 7.5, 150 mM NaCl, 1% IGEPAL-CA630, 0.5% Sodium Deoxycholate) supplemented with the following DUBs inhibitors: 20 mM of N-Ethylmaleimide (Sigma, cat#E3876-5G), 20 µM PR-619 (Life-Sensors, cat#SI-43265) and 5 µM of 1,10-phenanthroline (Life-Sensors, cat# SI-9649-0500). Protein lysates were denatured by heating for 10 min at 95 °C in presence of 1% SDS and next quenched by 10x dilution in ubiquitin buffer (50 mM Tris-HCl, 250 mM NaCl, 0.1% Triton X-100, 1 mM EDTA, 1 mM DTT). Myc-hnRNPK was captured during 2 h incubation with Protein A Sepharose CL-4B beads (GE Healthcare) previously coated with anti-myc tag antibody. Finally, immunoprecipitated samples were resolved on a 10% SDS-PAGE acrylamide gel. Polyubiquitylated hnRNPK was detected by immunoblot with either anti-HA or His antibody.

## Immunohistochemical staining

Tissue slides were deparaffinized by a 30 min incubation at 55 °C followed by hydration through xylene and clearing with 100–30% EtOH gradient. Antigen unmasking was carried out by 10 min boiling of slides in Citra Plus Antigen Retrieval buffer (Biogenex Laboratories). To inhibit endogenous peroxidase activity slides were incubated 10 min in 3% $H_2O_2$. Blocking of non-specific binding sites was performed for 10 min with buffered casein solution (Power Block Reagent, Biogenex). Primary antibody and biotinylated secondary antibodies were diluted in 5% goat serum/PBS solution and incubated with tissue samples for 2 and 1 h, respectively. Avidin-HRP complexes were conjugated to biotinylated secondary antibody using VECTASTAIN Elite ABC Kit (Vector Laboratories, cat#PK-6100). The labeling was performed by DAB (3,3'-diaminobenzidine) reaction applying DAB Substrate Kit (Vector Laboratories, cat#SK-4100). Before coverslip mounting, slides were counterstained with hematoxylin for 30 s, dehydrated and cleared in xylene.

## Immunofluorescence staining and quantification

c-Myc was detected on melanoma tissue microarray with the use of an Alexa488-conjugated c-Myc antibody. Tissue slides were deparaffinized during 30 min incubation at 55 °C followed by hydration through xylene and clearing with 100–30% EtOH gradient. Antigen unmasking was carried out by boiling the slide for 10 min in Citra Plus Antigen Retrieval buffer (Biogenex Laboratories). Blocking of non-specific binding sites was performed for 30 min with buffered casein solution (Power Block Reagent, Biogenex). Alexa488-c-Myc antibody was diluted in 5% goat serum/PBS solution and incubated with the tissue slide for 2 h. Coverslip was mounted with ProLong Gold antifade reagent (Invitrogen, cat#P36931) containing DAPI for nuclei counterstaining. Detection and quantification of the nuclear c-Myc fraction was performed with ImageJ software using an automatic algorithm to generate a nuclear outline based on the DAPI channel and overlay of the outline onto the Alexa488 channel to measure the mean intensity of Alexa488 fluorescence from 200 randomly picked nuclei/individual. Data processing was performed with the use of Phyton 3.10.

## IRES luciferase reporter assay

NIH3T3 cells were transfected by GenJet (SignaGen) with 0.6 µg reporter vector and depending on the condition 0.6 µg of myc-hnRNPK or/and Flag-Fbxo4. Twenty-four hours post-transfection, cells were plated at $8 \times 10^4$/well in a 48-well plate and lysed 24 h after plating for signal evaluation. Luc-Pair Duo-Luciferase Assay Kit 2.0 was applied for lysis and signal induction (GeneCopoeia, cat#LF001). Luminescence was detected on Varioscan Lux plate reader (ThermoFisher).

## RNA cross-linking assay

hnRNPK was ubiquitylated in vivo in HEK293T cells according to described above protocol. Forty-eight hours post-transfection, cells were irradiated twice—333 and 140 mJ/cm$^2$ in order to cross-link RBPs and RNA (A non-crosslinked cells were used as a control). Upon collection, cells were lysed in a buffer containing 0.1% SDS, 1%Triton X-100, 20 mM Tris-Cl pH=8.0, 150 mM NaCl and protease inhibitor (Roche cOmplete mini, cat#11836153001). In further steps, lysate was treated with RQ1 DNAse (Promega, cat#M6101) for 5 min at 37 °C at 1000 rpm (shaking) and subsequently with high (1:100) or low (1:10,000) concentration of RNAse I (Ambion cat# AM2295) to be finally cleared by 15 min centrifugation at $21,000 \times g$. Myc-hnRNPK was captured by incubation with protein A Dynabeads (Invitrogen, cat# 10002D) and 2 µg of anti-myc tag antibody in presence of 200 U RNAse inhibitor (Invitrogen, SUPERase, cat#AM2694) at 4 °C, overnight. After immunoprecipitation, hnRNPK-RNA complexes were washed, and

subjected to dephosphorylation with 1 µl of T4 PNK Enzyme (NEB, cat#M0201L) for 30 min at 37 °C, 1000 rpm horizontal shaking. Beads were washed 4 times with PNK buffer to next perform radiolabeling of RNA with 2 µl of T4 PNK enzyme and $P^{32}$-γ-ATP incubated for 60 min at 37 °C with 1000 rpm rotation. All samples were resolved by electrophoresis with the use of Novex NuPAGE system and were transferred onto nitrocellulose membrane which was subjected to autoradiographic visualization of signal.

## Ribosome profiling

Cells were expanded to appropriate amount necessary for the library generation for Ribo-seq and parallel RNA-seq. The cells were lysed on ice using 1.5 volumes of lysis buffer per gram of cells (dry weight). The appropriate volume of cell extract was treated with MNase, $CaCl_2$ (5 mM final concentration), and Turbo DNAse I and incubated at 25 °C for 30 min; the digestion was stopped with SUPERase-inhibitor and placing on ice. The cell extract is then loaded onto a 15–45% sucrose gradient to separate the polysomes by ultra-centrifugation. The relevant fractions corresponding to the 80S monosome were collected and precipitated using ethanol and the RNA was isolated. In parallel, total RNA was isolated from each sample and subjected to the Turbo DNase I digestion and then RNA-seq libraries were constructed using Illumina's TruSeq Stranded Total RNA kit. The ribosome-protected fragments were subjected to 15% denaturing polyacrylamide-mediated gel electrophoresis and were gel-extracted based on size (26-mers to 34-mers). The RNA was precipitated and subjected to the RiboMinus Ribosomal RNA depletion kit (Ambion). The rRNA-depleted RNA footprints were then subjected to de-phosphorylation. Adapter ligation, reverse transcription, and subsequent cDNA amplification to introduce library barcodes and UMI's (unique molecular identifier) was performed using the NEXT-Flex smRNA-Seq Kit v3 (Perkin Elmer); the products were purified using 8% native gel electrophoresis and subsequent gel extraction for the final library preparation.

## Invasion assay

Invasion of MEFs was determined by Cell Invasion Assay Kit (Sigma-Aldrich, cat#ECM550) while CytoSelect Cell Invasion Assay (Cell Biolabs, cat#CBA-110) was applied for melanoma cell lines. In detail, cells were transfected with control or anti-hnRNPK siRNA, 48 h post-transfection cells were serum starved for 24 h then $5 \times 10^5$ cells were resuspended in serum-free media and transferred into the upper part of a Boyden chamber pre-coated with extracellular matrix while the bottom part contained 10% FBS as a chemoattractant. Cells were incubated 24–72 h and stained with provided cell stain solution.

## Cell tracking assay

In all, 6000 cells/well were seeded in a 12-well plate. Sixteen hours after inoculation, cell motility was assessed on an automated stage microscope for 10 h with 20 min intervals for picture capturing. Cell tracking was performed with ImageJ manual tracking tool. Obtained results were processed and visualized using Chemotaxis and Migration Tool (iBiDi, https://ibidi.com) with the $X/Y$ calibration = 1.09 µm/pixel (adjusted to the microscope)

## Cell doubling time evaluation

In all, $1 \times 10^4$ MEF$^{wt}$ or MEF$^{Fbxo4-/-}$ cells were seeded in 6-well plates in triplicate for each condition. Every 48 h, cells were detached by trypsinization and counted with a hemocytometer. The cell death rate was evaluated by trypan blue (Sigma, cat#T8154) exclusion. Cells were monitored up to 8 days and results were plotted as a growth curve.

## Amino acid sequence alignment

The Lalign tool[51] was utilized to find non-overlapping local alignments between amino acid sequence of cyclin D1 and hnRNPK. Full output of analysis is attached in Supplementary Data 2.

## Nucleocytoplasmic fractionation

Cell were gently harvested in PBS. Obtained pellets were lysed in 6 volumes of buffer containing 10 mM Hepes (pH = 7.9), 50 mM NaCl, 0.5 M Sucrose, 0.1 mM EDTA and 0.5% Triton X-100 supplemented with protease inhibitors (PMSF, Aprotinin) and phosphatase inhibitors (NaF, β-glycerophosphate, $Na_3VO_4$). After 5 min incubation with lysis buffer, samples were centrifuged at $200 \times g$ at 4 °C to pellet nuclei. Supernatant was cleared at $18,000 \times g$, 4 °C and saved for western blot while nuclear fraction was first suspended in washing buffer (10 mM HEPES (pH = 7.9), 10 mM KCl, 0.1 mM EDTA and 0.1 mM EGTA), pellet at $200 \times g$ centrifugation. Finally, nuclear pellets were lysed in 4 volumes of buffer containing 10 mM HEPES (pH = 7.9), 0.5 M NaCl, 0.1 mM EDTA and 0.1 mM EGTA, 0.1% IGEPA-CA630 by 15 min medium-speed vortexing 4 °C followed by 10 min centrifugation at $18,000 \times g$, 4 °C.

## AHA labeling and Click chemistry

Click chemistry was applied to monitor the dynamics of c-Myc synthesis. This method exploits the ability of L-azidohomoalanine (AHA) to be incorporated in protein during translation as a substitute of methionine. Incorporated AHA is ligated to the alkyne-containing fluorophore–TAMRA in a click reaction. Experiment was performed as follows: HEK293T cells were incubated in methionine deficient DMEM media for 2 h. Next, cell were either harvested (0 time point) or treated with 50 mM of AHA for 15 or 30 min. Cell pellets were lysed in NP40 buffer (50 mM Tris pH = 7.5, 150 mM NaCl, 1% IGEPAL-CA630, 0.5% Sodium Deoxycholate) and cleared by $18,000 \times g$ centrifugation at 4 °C. A total of 300 µg of obtained lysate was used for in vivo ubiquitylation assay (see methods in vivo ubiquitylation assay) under native conditions; 500 µg of lysate was incubated with protein A Sepharose beads precoated with c-Myc antibody. After c-Myc precipitation, beads were washed 5 times with lysis buffer and boiled for 5 min in buffer containing 1% SDS, 50 mM tris (pH = 8) in order to disassociate precipitates from the beads. Next, samples were transfer to reaction soup containing 4uM of TAMRA (Thermofisher cat#T10183) and components from Click-iT Protein Reaction Buffer Kit (Invitrogen, cat#C10276), reaction was performed as indicated in the Click-iT kit manual. Newly synthesized c-Myc was detected by SDS-PAGE gel resolution followed by anti-TAMRA western blot.

## Cancer databases

Analysis of Fxr1 and hnRNPK expression presented in Fig. 6B and S6G were generated by using Gepia2 tool (http://gepia2.cancer-pku.cn) based on TCGA and GTEx resource. Copy number variation for *CRYAB* gene presented in Fig. 6C were generated on NIH Genomic Data Commons Portal v1.28.0 (https://portal.gdc.cancer.gov/). The expression and copy number variation of Fxr1 in esophageal cancer presented in Fig. 6H, I was acquired from Oncomine database (https://www.oncomine.org) utilizing the following subset of data Su- Esophagus2, Hu- Esophagus. The survival analysis tool at The Cancer Proteome Atlas (https://tcpaportal.org) was utilized to acquire Kaplan–Maier plot presented in Fig. S6E.

## Motif abundance analysis

oRNAment tool (http://rnabiology.ircm.qc.ca/oRNAment) was used to determine RPB putative binding site instances in the group of Fbxo4-responsive genes. Algorithm recognized 95.89% of input. Input list and, non-recognized genes available in Supplementary Data 3.

## GSEA analysis

All GSEA analysis were run on the 4.2.1 software version (https://www.gsea-msigdb.org) with the following settings: Number of permutations: 1000; gene symbol: collapse; permutation type: gene set, Chip platform: Mouse_ENSEMBL_Gene_ID_Human_Orthologs_MSigDB.v.7.5.1.chip. GSEA input is presented in Supplementary Data 4. The list of c-Myc

downstream effectors were generated based on the molecular signatures from GSEA database with the following systematic name: M2477, M15774, M17557, M6506, M2310, M18501, M6951, M2607, M3456, M2069, M66, M139, M1756, M3923, M1249, M17753, M27, M5926, M5928

## RNA sequencing

WM983B melanoma cell lines were infected with lentivirus encoding anti-Fbxo4 shRNA or empty vector. 72 h post-infection cells went through negative selection by 0.5 μg/ml puromycin treatment. In all, $2.5 \times 10^6$ cells were plated 24 h prior to harvesting for RNA purification using RNeasy kit (Qiagen, cat# 74104) while 10% of cells were lysed and subjected to western blot analysis in order to confirm hnRNPK knockdown. Obtained RNA was subjected to sequencing (RNA-seq) and bioinformatic analysis by Novogene Co., LTD (Beijing, China). The results with adjusted $p$ value <0.05 were considered as a significant.

## Evaluation of c-Myc synthesis rate

In all, $3 \times 10^6$ MEF$^{wt}$ or MEF$^{Fbxo4-/-}$ were seeded 24 h before 120 min treatment with 100 μg/ml cycloheximide to deplete c-Myc protein. Next, cells were washed twice with 10 ml warm DPBS and re-feed with DMEM containing MG132 (working concentration 10μM) to prevent proteasome dependent degradation of newly synthesized c-Myc. Cells were harvested at 0 (no refeed), 10, 20, 30, and 45 min to be lysed in EBC buffer (50 mM Tris-Cl pH = 8, 120 mM NaCl, 0.5% IGEPA-CA630) and cleared by 15 min, $21,000 \times g$ centrifugation. Finally, 40 μg of lysate was resolved on 10% SDS-PAGE gel and transferred to PVDF membrane (3 h, 40 V).

## Bioinformatic analysis

The ribosome profiling libraries underwent single-end sequencing. The resulting Fastq files were processed using FastQC (v0.11.8; RRID:SCR_014583) for QC analysis. The preprocessing was performed based on the QC report using FASTQ Quality Filter module in the FASTX-Toolkit (RRID:SCR_019035) which was used to extract the bases with 99% accuracy based on Q Score. Reads where less than 70% of bases had an accuracy of at least 99%, were removed. The sequences were then collapsed by UMI and Cutadapt (RRID:SCR_011841) was performed to trim 21-nt adaptor before the first and the last 4-nt were trimmed from the reads to remove the UMI. The reads were aligned to mm10 (Genome Reference Consortium Mouse Build 38 (GCA_000001635.2)) using STAR v2.5.3a (RRID:SCR_004463). The generated bam files were processed with RiboProfiling package v1.2.2[52] and the coverage counts on the coding regions (CDS) were obtained for each sample based on RiboProfiling function modules, TxDb.Mmusculus.UCSC.mm10.knownGene v3.10.0 (Annotation package for TxDb object(s)) and GenomicFeatures package (v1.46.1; ref. [53]) DESeq2 pipeline (version 1.26.0) was performed based on the expressed raw reads for the differential express gene analysis. The specific thresholds for the features in the comparison result such as adjusted $p$ values and log2 fold changes were set up for the gene extraction[54]. Replicate correlation and normalized read counts per mRNA feature and are presented in Figs. S8 and S9, respectively. To calculate a $p$ value for ribosome accumulation (Fig. 3G) IntersectBed was used to determine the number of unique reads mapped to the 25nt surrounding nucleotide 442 of the c-myc transcript for all samples and their replicates. These $p$ values were normalized to the total number of reads in per library before subjecting them to a Student's $t$ test.

## Statistical analysis and reproducibility

Data were collected and handled in either GraphPad 8 or Microsoft Excel. Statistical analysis: two-tailed Student's $t$ test (unpaired or ratio paired), Mann–Whitney $U$-test, or two-way Anova were computed in GraphPad 8. All data with $p < 0.05$ was considered statistically significant. Western blot experiments presented in Figs. 2H, D; S3G, H, L; S4E; and S5B were performed in two biologically independent replicates. Western blot experiment presented in Figs. 1H; 2A–C, F, G; 3B, C, K, L; 4E; 5A; S1A, B, D, E; S2A, C, E–J; S3D, B, E; and S5A, C were performed in at least three biologically independent repeats. Rybo-seq was run in biological duplicates. Remaining experiments were performed in at least three biologically independent replicates.

## Reporting summary

Further information on research design is available in the Nature Research Reporting Summary linked to this article.

## Data availability

Total RNA sequencing data of melanoma cell line (WM983B) generated during this study is available at NCBI Gene Expression Repository: GSE196396. Ribosome profiling data of MEFs generated through the study is available at NCBI Gene Expression Repository: GSE196483. The following cancer databases has been used to collect the data with the use of code implemented in their online tools: Gepia2 tool (http://gepia2.cancer-pku.cn); NIH Genomic Data Commons Portal v1.28.0 (https://portal.gdc.cancer.gov/); Oncomine database (https://www.oncomine.org); The Cancer Proteome Atlas (https://tcpaportal.org); The Encyclopedia of RNA Interactomes (https://starbase.sysu.edu.cn/). All uncropped scans of blots are available with this article. The authors declare that all the data supporting the findings of this study are available within the article, its Supplementary Information files, and source data file. Source data are provided with this paper.

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

## Acknowledgements

Authors would like to thank Dr. Anne Catherine Prats for sharing IRES reporter constructs, Dr. Gauravi Deshpande from Cleveland Clinic for assistance in analysis of cell tracking experiments, Jianhua Hu from Biostatistics Core at Columbia University (NY) for the support in statistical analysis, Anna Mucha for preparing illustrations, Piotr Mucha for

informatic support, Marganit Farago-Tirosh and Polly Phillips-Mason for editorial work. This research was supported by National Cancer Institute: P01 CA098101 (J.A.D., A.K.R., A.J.B., K.-K.W., H.N.) and National Institutes of Health R01DK53307 and R01DK060596 (M.H.).

## Author contributions

Conceptualization: J.A.D. Methodology: B.M., J.A.D., E.J., F.T., M.H., S.Y.F., H.N. Formal analysis: B.M., Z.G., F.T., G.Z., S.Q., D.P., O.S.C., A.J.K.-S. Investigation: B.M., S.Q., V.T., S.B., Z.G., F.T., D.P., S.F., S.M., H.H., H.N. Resources: J.A.D, E.J., H.N., A.K.R., M.H., S.Y.F., I.M.,.A.B. Data curation: B.M., F.T., G.Z., E.J., J.N.D. Writing—original draft: J.A.D., B.M. Writing—review and editing: S.Y.F., E.J., M.H., A.B., K.K.W., A.J.B., A.K.R., C.J.D. Visualization: B.M., F.T., G.Z. Supervision: J.A.D. Funding acquisition: J.A.D., K.K.-W., A.B., A.K.R., H.N.

## Competing interests

The authors declare no competing interests.

## Additional information

[1]Department of Biochemistry, Case Comprehensive Cancer Center, Case Western Reserve University, Cleveland, OH 44106, USA. [2]Curriculum in Genetics and Molecular Biology, University of North Carolina at Chapel Hill, Chapel Hill, NC, USA. [3]Center for RNA Science and Therapeutics, Case Comprehensive Cancer Center, Case Western Reserve University, Cleveland, OH 44106, USA. [4]Department of Genetics and Genome Sciences, School of Medicine, Case Western Reserve University, Cleveland, OH 44016, USA. [5]Division of Hematology-Oncology, Department of Medicine, Herbert Irving Comprehensive Cancer Center, Columbia University Irving Medical Center, New York, NY 10032, USA. [6]Histopathology Facility, Fox Chase Cancer Center, Philadelphia, PA 19111, USA. [7]Department of Pathology and Laboratory Medicine, Medical University of South Carolina, Charleston, SC 29425, USA. [8]Department of Clinical Chemistry and Biochemistry, Medical University of Lodz, 60 Narutowicza St. 90-136, Lodz, Poland. [9]Lineberger Comprehensive Cancer Center, University of North Carolina at Chapel Hill, Chapel Hill, NC 27599, USA. [10]Department of Pharmacology, University of North Carolina at Chapel Hill, Chapel Hill, NC 27599, USA. [11]Division of Digestive and Liver Diseases, Department of Medicine, Herbert Irving Comprehensive Cancer Research Center, Columbia University Irving Medical Center, New York, NY 10032, USA. [12]Division of Hematology and Medical Oncology, Perlmutter Cancer Center, New York University, New York, NY 10016, USA. [13]Department of Biomedical Sciences, School of Veterinary Medicine, University of Pennsylvania, Philadelphia, PA 19104, USA. [14]Case Comprehensive Cancer Center, Case Western Reserve University, Cleveland, OH 44106, USA. ✉e-mail: jad283@case.edu

