## [Peer Review File · Nature Communications]

Tumor suppressor mediated ubiquitylation of hnRNPK is a barrier to oncogenic translationREVIEWER COMMENTS

Reviewer #1 (Remarks to the Author):

Mucha et al find a novel relationship between the (tumor suppressive) functions of SCF-Fbxo4, ubiquitylation of hnRNPK and translation of c-Myc. SCF-Fbxo4 leads to ubiquitylation of hnRNPK in residues K21/34, which in turn results in reduced binding of hnRNPK to c-Myc mRNA and reduced IRES-dependent c-Myc translation, reduced cell invasiveness, tumor growth and metastatic capacity of tumors. The study is quite complete, including molecular analysis, mice xenografts and histological analysis of patient samples in melanoma and ESCC. The claims on the specific mechanism of translational regulation of c-Myc by hnRNPK are not supported by strong, compelling data, but other than this I find the manuscript convincing provided that sufficient data are included and unclear statements (or figures) clarified as indicated below. I will mention these in order of appearance in the main text:

1) Why did the authors focus on hnRNPK, and not on the many other KH-domain containing proteins mentioned in Figure 1 (e.g. PCBP2)?

2) How many Ribo-Seq and RNA-seq biological replicates were performed? Please, show correlation and PCA analysis.

3) Figure 1C: On which RNA set was this analysis performed, on the 3649 mRNAs? Could the authors perform the same analysis on genes changing only at the level of translation? Is hnRNPK dominant in this case?

4) Figure 1E: the authors mention in Materials and Methods that these data were obtained from an extended analysis on material acquired previously. Extended in which way? Please, briefly explain. How many independent replicates were performed? Please, include an excel table with the full interactome showing coverage, areas for the different conditions used including negative controls, and p-values or FDRs.

5) Figure S1D: The authors mention that hnRNPK binds to the C-terminus of Fbxo4, but deletion of the N-terminal domain also results in loss of hnRNPK binding.

6) Figure 3D-E: Please, include an excel table with the 700 SCF-Fbxo4/hnRNPK downstream targets, indicating which ones are c-Myc targets and showing the level of downregulation by loss of Fbxo2 as well as % rescue upon hnRNPK KD. Indicate in the figure legend the number of independent replicates for RNA-Seq and ribosome profiling for each condition, and show the correlation between replicates and PCA analysis in a Supplementary figure.

7) Figure 3G: Please, show the position of the AUG in the X-axis. Also show basic controls of ribosome profiling, such as trinucleotide periodicity and reduced ribosome density in UTRs. Show this periodicity especially at the claimed paused site. How reproducible is ribosome pausing at this site? Is this site rich in unusual codons? How is pausing at a site within the ORF (i.e. defects in elongation) reconciled with defects in IRES-mediated translation of a reporter whose ORF has nothing to do with that of c-Myc (i.e. defects in initiation)? Why pausing at this site is alleviated by non-ubiquitylated hnRNPK? Altogether, the data on the mechanism of translational repression is too weak, especially regarding potential defects in elongation. Should no further data supporting this claim be added, Figure 3G should be deleted and the discussion changed accordingly.

8) Figure S3L: I do not understand this figure. If the left panels represent IP of hnRNPK and labelling of bound RNA with PNK: What is the band detected in the non cross-linked samples? Is it hnRNPK? If so, and given that this is a denaturing gel that should separate protein from RNA, how come one detects radioactivity (that is, RNA bound to hnRNPK) here? On the same line, in the crosslinked samples upon treatment with high RNase concentration (H lanes), one should see a band corresponding to hnRNPK, which should be weaker upon hnRNPK ubiquitylation. In addition, as control these same membranes should be subjected to WB against hnRNPK to ensure equivalent IP of the protein. Further, I do not see what the right panel represents. Please, explain this figure in more detail in the Figure legend and main text.

9) Figure 4C-D: Why is there no difference in invasiveness by depletion of hnRNPK in wt MEFs? According to Figure 3, such depletion leads to reduced c-Myc expression.

10) Figure S5A: Please, show a better blot. No interaction can be seen here even for wt Fbxo4.

11) Figure S5D-I: What do the authors conclude from these panels? WM983B cells are Fbxo-wt and their invasiveness is not affected by depletion of hnRNPK, yet genes involved in ECM remodeling, etc, are affected by hnRNPK depletion.

12) Figure 5E: Can the authors show images with similar background levels? The image corresponding to the vector seems to have been exposed longer, as the whole lung is visible. Same for Fig S6B.

13) Figure 5I-K: I don't see differences between vehicle and JQ1 treatment, as the authors claim.

14) Fig 5M: It is surprising to see such a difference in Myc staining at 40x when barely no differences are observed at a lower magnification. Please, quantify.

15) Figure 6F: Can the authors show a higher magnification of this image to clearly visualize cytoplasmic accumulation of hnRNPK? Same for Figure 6K.

Other comments:

- Fig 1 legend: F, please indicate what does the asterisk mean. Please, position G, H section explanations correctly.
- Fig S1B: Please, explain what ΔF is at this point or eliminate from this figure panel.
- Fig S1C: Please, spell-out the domains of both proteins and explain their roles in the legend.
- Lane 116: I guess the authors refer to Fig S2B (not S1B)
- Fig S2C: Please show more contrasted panels for hnRNPK. The protein is barely visible in several cell lines.
- Figure S2F: There is almost no difference in the ubiquitylation of hnRNPK between Fbxo WT and delta-F. Please, quantify, correct for input hnRNPK and indicate at bottom of WB.
- Figure S2H: Explanation missing in Figure legend.
- Lane 147: S3C (not SC3)
- Figure S2I is not mentioned in the main text
- Figure S3D-E: I don't see much of a rescue of Myc expression by over-expression of ubiquitin-refractory hnRNPK, as mentioned in the main text. Please, quantify, correct for hnRNPK expression (myc-K), and indicate at bottom of WB.
- Figure 3E: Scatter plots for hnRNPK KD section are switched.
- Lane 189: Figure 3K addresses the role of Fbxo4 in promoting the synthesis of endogenous c-Myc, but says nothing about hnRNPK. Please, eliminate 'hnRNPK' from the sentence.
- Lane 204: I guess the authors refer to Fig S3M (not Fig 3).
- Lane 2013: Fig 4A,B represents the analysis of the 700 Fbxo4-hnRNPK targets? If so, please indicate clearly by adding the number, as right now one could think that any target deemed as dysregulated in Fig 3E counts. If not, please show the analysis of the 700 targets.
- What is the status of endogenous Fbxo4 in B16F10 cells? Is it wt or mutated?
- Lane 287: I guess the authors refer to translational control (not post-translational).
- Lane 294: I guess the authors refer to Fig 6F (not 6G).
- Figure 6H: What is the difference between the right and left panels?

Reviewer #2 (Remarks to the Author):

In this manuscript, Mucha et al. describe a role of RBP hnRNPK in regulation of translation, in particular of the proto-oncogene c-MYC, which promotes hnRNPK function in tumorigenesis. The authors suggest that the oncogenic function of hnRNPK depends on its cytoplasmic function that is regulated by ubiquitylation. Ubiquitylation is proposed to limit the ability of hnRNPK to bind to its

target mRNA including c-MYC RNA.

My review is focused on the experiments that establish the E3 ligase-substrate relation between hnRNPk and SCF-Fbxo4. The authors show relatively convincingly that hnRNPk can be ubiquitylated by Fbxo4 in vitro, however the in vivo evidence remains scarce and not convincing. Also, that hnRNPk is modified by K63-linked ubiquitylation is not convincing. In general, I have problems with the quality of the western blots that demonstrate the interaction between hnRNPk and Fbxo4 as well as ubiquitylation of hnRNPk in vitro and in vivo (especially the ones shown in extended data).

Figure 1H demonstrates the interaction between hnRNPk and Fbxo4. I appreciate that this is an IP with endogenously expressed proteins, however the band with hnRNPk is hardly visible. This IP should be done in more replicates and preferably with quantification and also blotting for a positive control such as CCND1.

Two panels Fig. 2F and G show evidence for in vivo ubiquitylation of hnRNPk, however the ubiquitylation pattern looks completely different. I could not find any information about the cell lines used for these assays and how the distribution of endogenous hnRNPk looks in these cell model and whether this localization pattern is resembled when using tagged hnRNPk. This is essential if the authors want to make claims about the cytoplasmic vs nuclear function and use both endogenous and overexpressed hnRNPk. The authors claim that hnRNPk is modified by K63-linked chains (which is atypical for Fbxo4) by using ubiquitin mutants but never test whether ubiquitylation of hnRNPk is affected upon proteasome inhibition. I would expect that it is not if these are indeed K63-linked chains. As it is now, these panels are meaningful without knocking down hnRNPk and showing that these smears are really on hnRNPk (especially since two completely different patterns of ubiquitylation are observed). Can the authors try K48 vs K63 Ub-specific antibodies?

Fig 2C shows in vitro ub of hnRNPk by either K48 or K63-linked chains. I find it surprising that the chain pattern look exactly the same on the WB except that K48 is less abundant. This would suggest that the authors look at the same chain type but in one case hnRNPk is less efficiently ubiquitylated.

Extended data on hnRNPk ubiquitylation are very confusing and most of the WB in extended Figure 1 and 2 do not seem to convincingly support the claims.

For example:

Fig.S1A – Fbxo4 does not seem to be enriched in hnRNPk IP.

Fig. S1B – hnRNPk is enriched in WT and deltaF Fbxo4 and this is not the case in main figure. There also seems to be a strong double band in the delta F mutant that resembles the size of the WT protein.

Fig. S1D – Fbxo4 is barely seen in the WT lane and seems that none of the mutants bind to hnRNPk. What is the explanation for that if only one terminus is important for binding?

Fig. S1E – Again the WT band is barely visible and the delta N mutant binds stronger than the WT. Is there any explanation for this?

In Figure S2 the authors show the dependency of ubiquitylation on K21 and K34 in hnRNPk. However, in panel E the ubiquitylation is completely gone in single K34R mutant and not really affected by K21R mutant. This seems to be then opposite in panel F where K34R barely affects ubiquitylation.

None of the CHX chase assays (either for MYC or hnRNPk) are accompanied by quantification or determination of half-lives. This is also true for all other WB. This should be at very least provided for key data where the representative WBs do not show a clear result.

The authors use the previously published MS interactome to show that hnRNPk is a putative substrate of Fbxo7 and this is only indicated in Methods section. I find this misleading. Also, no information about the number of replicates and statistical analysis is provided. Simply providing a table with selected proteins and number of identified peptides is not acceptable. I suggest that the authors completely remove the previously published interactome analysis from the figure and simply mention that hnRNPk was identified as putative substrate in a previous study with indicated reference.

MLN4924 is misspelled in Fig. 1A.

The figure legends are in general too briefly explained without providing any details on number of replicates and statistics.

There is no legend for figure S2H.

Reviewer #3 (Remarks to the Author):

In this manuscript, Diehl and co-workers report a molecular mechanism by which the ubiquitin pathway restricts oncogenic activity of hnRNPk. They identified hnRNPk as a target of the ubiquitin ligase SCF-Fbxo4, which ubiquitylates hnRNPk through K63 linkages rather than the degradative K48 linkages. By using a genome-wide screening, they identified c-Myc as a target of SCF-Fbxo4-hnRNPk. Loss of Fbxo4 triggers hnRNPk-dependent increasing in c-Myc translation and tumorigenesis. Taken together, the work by Mucha et al. demonstrated an essential role of the ubiquitin ligase Fbxo4 in limiting hnRNPk function to maintain appropriate translational homeostasis.

Overall, this is a well-written manuscript, demonstrating an important role for a ubiquitin ligase in c-Myc translational regulation and limiting tumorigenesis. The genome-wide screenings and analyses are convincing, and the functional significance of the Fbxo4-hnRNPk in vivo and in human subjects is clearly shown. I have only a few concerns regarding the molecular mechanism of this regulatory axis, as described below.

1) The authors tried to demonstrate in vitro that SCF-Fbxo4 ubiquitylates hnRNPk through K63-linkages. However, the ubiquitin chain specificity generated by the RING-type E3s depend on the property of the E2 enzyme. As UBC5 is a promiscuous E2, it can assemble ubiquitin chains either through K48, K63, or K11 linkages. More convincing evidence in cell is needed.

2) It is unclear from the manuscript how the K63-linked ubiquitylation affects the function of hnRNPk.

3) In Fig. 3B and Fig. S3D, a portion of Fbxo4 remains in the Fbxo4 ^{-/-} cells, while hnRNPk is completely lost in the knockdown cells. Please explain this. I guess they might be mislabeled?

4) In Fig. 2G, the anti-HA blot is of poor quality. I suggest that anti-ubiquitin blot of the same IP may improve the quality.

Minor comment

Fig. 3K: Mg132 should be corrected to MG132.

Reviewer #1 (Remarks to the Author):

Mucha et al find a novel relationship between the (tumor suppressive) functions of SCF-Fbxo4, ubiquitylation of hnRNPK and translation of c-Myc. SCF-Fbxo4 leads to ubiquitylation of hnRNPK in residues K21/34, which in turn results in reduced binding of hnRNPK to c-Myc mRNA and reduced IRES-dependent c-Myc translation, reduced cell invasiveness, tumor growth and metastatic capacity of tumors. The study is quite complete, including molecular analysis, mice xenografts and histological analysis of patient samples in melanoma and ESCC. The claims on the specific mechanism of translational regulation of c-Myc by hnRNPK are not supported by strong, compelling data, but other than this I find the manuscript convincing provided that sufficient data are included and unclear statements (or figures) clarified as indicated below. I will mention these in order of appearance in the main text:

1) Why did the authors focus on hnRNPK, and not on the many other KH-domain containing proteins mentioned in Figure 1 (e.g. PCBP2)?

In the preliminary co-IP and ubiquitylation assays hnRNPK presented the highest affinity to Fbxo4. In addition, there is a substantial base of clinical reports indicating hnRNPK dysregulation in cancer.

2) How many Ribo-Seq and RNA-seq biological replicates were performed? Please, show correlation and PCA analysis.

We have added the requested figure to the supplementary data.

Figure S8 | Replicate Correlation. A) RNA-seq library heatmap of Z-score correlations of each condition with replicates. The scale blue to white indicated increasing similarity, with white being perfect correlation. B) RNA-seq library Principal Component Analysis (PCA) of each condition with replicates. C) Ribo-seq library heatmap of z-score correlations of each condition with replicates. D) Ribo-seq library Principal Component Analysis (PCA) of each condition with replicates.

3) Figure 1C: On which RNA set was this analysis performed, on the 3649 mRNAs? Could the authors perform the same analysis on genes changing only at the level of translation? Is hnRNPK dominant in this case?

95.89 % of gene IDs used as input was recognized by oRNament algorithm therefore precise number of targets included in the analysis is 3499. oRNAMENT analysis on targets responding at the level of RPF has been run and show the same (3rd) position of hnRNPK in terms of motif abundance (208709 hits) within KH domain RBPs.

4) Figure 1E: the authors mention in Materials and Methods that these data were obtained from an extended analysis on material acquired previously. Extended in which way? Please, briefly explain. How many independent replicates were performed? Please, include an excel table with the full interactome showing coverage, areas for the different conditions used including negative controls, and p-values or FDRs.

Mass-spectrometry was performed on single samples of material co-immunoprecipitating with ectopic wt Fbxo4 or ectopic Fbxo4 ΔF expressed in Fbxo4 null MEFs treated with DMSO or proteasome inhibitor (MG132). Prominent bands were identified by silver staining and excised for analysis. Extended analysis refers to analysis of bands other than those indicated in our previous publication (1). To provide transparency to this point and take into consideration other reviewer comments, we removed the mass spectrometry procedure from material and methods and quote our previous article as the source of the data.

1- Qie, Shuo, et al. "Fbxo4-mediated degradation of Fxr1 suppresses tumorigenesis in head and neck squamous cell carcinoma." Nature communications 8.1 (2017): 1-14.

Body text of the manuscript has been modified in by replacement of the sentence:

We also assessed binding of RBPs to Fbxo4 via its purification followed by mass spectrometry analysis. Multiple KH-domain RBPs co-purified with Fbxo4 with the highest peptide coverage for hnRNPK (Fig 1E).

By the following sentence (lane 93):

Analyzing material collected during a previous purification/mass spectrometry experiment in which Fxr1 was identified as a Fbxo4 target (21), we noted multiple KH-domain RBPs that co-purified with Fbxo4 with hnRNPK as one of high potential relevance (Fig 1E).

5) Figure S1D: The authors mention that hnRNPk binds to the C-terminus of Fbxo4, but deletion of the N-terminal domain also results in loss of hnRNPk binding.

Functional/active SCF-Fbxo4 requires dimerization mediated by the N-terminal D-Domain (1).

Therefore, a non-dimerizing mutant is expected to present impaired substrate recognition.

1- Barbash, Olena, et al. "Mutations in Fbx4 inhibit dimerization of the SCFFbx4 ligase and contribute to cyclin D1 overexpression in human cancer." *Cancer cell* 14.1 (2008): 68-78.

6) Figure 3D-E: Please, include an excel table with the 700 SCF-Fbxo4/hnRNPk downstream targets, indicating which ones are c-Myc targets and showing the level of downregulation by loss of Fbxo2 as well as % rescue upon hnRNPk KD. Indicate in the figure legend the number of independent replicates for RNA-Seq and ribosome profiling for each condition, and show the correlation between replicates and PCA analysis in a Supplementary figure.

A spreadsheet with targets presented in Fig3D-E has been generated and attached to Supplementary Data 1. PCA and correlation has been presented as requests in comment 2. Figure legend was adjusted by adding the following statement (lane 990):

Data presented in the A,D,E,F,G is based on the RNA-seq and Ribo-seq data run in biological duplicates. Correlation between replicates and PCA analysis of RNA and Ribo-seq is presented in Fig S8. Detailed list of targets, fold change and rescue levels presented in (E) is available in Supplementary Data1

7) Figure 3G: Please, show the position of the AUG in the X-axis. Also show basic controls of ribosome profiling, such as trinucleotide periodicity and reduced ribosome density in UTRs. Show this periodicity especially at the claimed paused site. How reproducible is ribosome pausing at this site? Is this site rich in unusual codons? How is pausing at a site within the ORF (i.e. defects in elongation) reconciled with defects in IRES-mediated translation of a reporter whose ORF has nothing to do with that of c-Myc (i.e. defects in initiation)? Why pausing at this site is alleviated by non-ubiquitylated hnRNPk? Altogether, the data on the mechanism of translational repression is too weak, especially regarding potential defects in elongation. Should no further data supporting this claim be added, Figure 3G should be deleted and the discussion changed accordingly.

We have revised Fig 3G according to the comment and added start and stop codons as well as the precise nucleotide number of the peak. We have also added a supplementary figure (**Figure S9**) to show the reduced ribosome protected fragments (normalized for RNA feature length) in the UTRs, compared to the ORF.

Periodicity is not readily extracted from the ribosome profiling data, because Micrococcal nuclease (MNase) was used to isolate the ribosome bound fragments. As is well known, MNase cuts with a pronounced base bias, and periodicity can't be inferred for larger transcriptomes, unless unrealistic sequencing depth is reached. MNase was used in the experiments, because mammalian ribosomes are much less susceptible to degradation by MNase than to degradation by RNase I, that is often used in experiments aiming for a determination of periodicity. In contrast to yeast ribosomes, mammalian ribosomes are readily degraded by RNase I, and meaningful measurements of ribosome occupancy and translation status require large, and in our case unrealistic amounts of cellular material. We emphasize, however, that accurate measurements of ribosome occupancy and translation status are achieved with MNase, as evidenced by numerous prior publications.

The pausing within the ORF (i.e. defects in elongation) is independent of the impact on IRES-mediated translation. These are two distinct effects. However, pausing in the ORF explains the reduction of protein production despite high ribosome occupancy within the ORF.

The alleviation of pausing at this site by non-ubiquitylated hnRNPk is likely due to change in RNA binding ability of hnRNPk, which presumably alters structure of the mRNA, changes in the

interaction of the RNA with other RNA binding proteins, or a combination of both. We agree with the reviewer that the data do not comprehensively explain how hnRNPK ubiquitylation and ribosome pausing are mechanistically linked. However, we feel that the data are important to report here, because they reconcile the marked ribosome occupancy of the c-MYC transcript with the reduction in protein production. In the absence of these data, this observation would lack context. For this reason, we prefer to keep the data panel in the figure.

Position of start and stop codon has been presented:

8) Figure 3S3L: I do not understand this figure. If the left panels represent IP of hnRNPK and labelling of bound RNA with PNK: What is the band detected in the non cross-linked samples? Is it hnRNPK? If so, and given that this is a denaturing gel that should separate protein from RNA, how come one detects radioactivity (that is, RNA bound to hnRNPK) here? On the same line, in the crosslinked samples upon treatment with high RNase concentration (H lanes), one should see a band corresponding to hnRNPK, which should be weaker upon hnRNPK ubiquitylation. In addition, as control these same membranes should be subjected to WB against hnRNPK to ensure equivalent IP of the protein. Further, I do not see what the right panel represents. Please, explain this figure in more detail in the Figure legend and main text.

We appreciate the reviewer's comment and agree that the previous figure was difficult interpret. To better explain this experiment and the obtained data, we have amended the figure by adding an

experimental scheme and by re-arranging the data panels in a more logical manner. We have also amended the text with a more detailed explanation of the experiment.

We examined binding of ubiquitylated and non-ubiquitylated hnRNPk to RNA in cells, using UV-mediated cross-linking followed by immunoprecipitation of hnRNPk and RNase digestion. This approach is identical to the workflow used for the CLIP (Cross-Linking with ImmunoPrecipitation) technique, a method that we and others have extensively used to map RNA-protein contacts in cells. Here, we omitted the mapping step (most frequently performed by Next Generation Sequencing), because our goal was to qualitatively show RNA binding to hnRNPk in the cell.

We performed the experiment as follows. Polyubiquitylation of hnRNPk was induced in HEK293T cells upon transfection with the expression vectors of Myc-hnRNPk, HA-ubiquitin and Flag-Fbxo4 or GFP for non-ubiquitylated control. Cells were next subjected to UV-induced RNA:protein crosslinking, which introduces a zero-length crosslink between protein and RNA and thus reports direct protein-RNA binding. We then harvested the cells and pulled down Myc-hnRNPk:RNA complexes. Next, RNA was radiolabeled with P³² using PNK. Of note, PNK is able to phosphorylate proteins as well, and this generates the well-defined bands in the non-crosslinked samples. RNA binding to hnRNPk leads to signal above hnRNPk, that due to the heterogenous nature of the bound RNAs appears as a “smear”. Of note, the “smear” is reduced in the samples treated with RNase, because the RNA is degraded. This is precisely seen in our samples.

We trust that the new figure and the text revisions constructively address the reviewer’s comment.

New figure Version

9) Figure 4C-D: Why is there no difference in invasiveness by depletion of hnRNPk in wt MEFs? According to Figure 3, such depletion leads to reduced c-Myc expression. Murine fibroblasts are commonly regarded as a non-invasive cells therefore loss-of invasion function by hnRNPk depletion is not expected in the wild type cells. We postulate that loss of Fbxo4 or overexpression of hnRNPk contribute to a gain of invasiveness phenotype that can be abrogated by hnRNPk depletion.

10) Figure S5A: Please, show a better blot. No interaction can be seen here even for wt Fbxo4.
 Panel has been replaced by higher quality data from another experiment

11) Figure S5D-I: What do the authors conclude from these panels? WM983B cells are Fbxo-wt and their invasiveness is not affected by depletion of hnRNPk, yet genes involved in ECM remodeling, etc, are affected by hnRNPk depletion.

Genomic data presented in the Fig S5D-I is analysis of WM983B cells transfected with anti-Fbxo4 RNAi vs. control RNAi, enrichment of motility/invasiveness related pathways due to loss of wt Fbxo4 is consistent with our hypothesis. Delivery of those informations has been enhanced by clear indication of genetic setup above each graph along with the correction in the results section.

D WM983B^{Fbxo4} siRNA vs. WM983B^{control} siRNA

E KEGG

F Reactome

G GO_MF

H GO_CC

I GO_BP

Body text correction (lane: 264)

Subsequently, we knocked down *Fbxo4* in WM983B to determine whether loss of *Fbxo4* would reproduce a similar genomic landscape to that observed in *Fbxo4*^{-/-} cells. Pathway analysis of RNA-seq revealed high representation of genes involved in ECM remodeling, focal adhesion, adherens junction, or integrin surface interaction (Fig S5D-I).

12) Figure 5E: Can the authors show images with similar background levels? The image corresponding to the vector seems to have been exposed longer, as the whole lung is visible. Same for Fig S6B.

All pictures were taken with a fixed 2 sec exposure time, auto-fluorescence background cut-off was set based on the signal from the non-transplanted mouse lung. Established filters were additionally validated by measurement on 3 additional non-injected mice as presented in S6B (bottom part). All original mixed and unmixed files along with spectral libraries (set background filtering) files are available. Contrast was adjusted to improve visibility.

New version

Old version:

13) Figure 5I-K: I don't see differences between vehicle and JQ1 treatment, as the authors claim. Better images are provided to illustrate the difference.

New

Old :

Panel J is quantification of K which was done based on measuring fluorescence intensity in 3 circular regions of the same area on each lung. Student t-test indicates a statistically significant difference.

14) Fig 5M: It is surprising to see such a difference in Myc staining at 40x when barely no differences are observed at a lower magnification. Please, quantify. Quantification of c-Myc signal intensity has been added

15) Figure 6F: Can the authors show a higher magnification of this image to clearly visualize cytoplasmic accumulation of hnRNPk? Same for Figure 6K.

Digital magnification of representative field has been provided next to the originally presented pictures.

Other comments:

• Fig 1 legend: F, please indicate what does the asterisk mean. Please, position G, H section explanations correctly.

Asterisk has been labeled. The position of G and H in the figure legend has been corrected.

F

	KH-domain RBPs	Motif abundance	Total count Norm.*
1	PCBP2	590 832	0.19
2	FUBP1	324 235	0.17
3	hnRNPk	250 524	0.19
4	IGF2BP2	249 178	0.17
5	PCBP1	242 114	0.19
6	KHDRBS2	226 300	0.17
7	IGF2BP3	222 523	0.17
8	Nova1	214 437	0.17

*- motif abundance normalized to the total count across the genome

• Fig S1B: Please, explain what DF is at this point or eliminate from this figure panel.

Explanation of ΔF relevance and function has been added to result section (see bolded):

Based upon these observations, we postulated that hnRNPk is regulated by SCF^{Fbxo4}. Consistently, endogenous (Fig 1H; S1A) and ectopic Fbxo4 (wt and catalytically deficient ΔF) coprecipitated with hnRNPk (Fig S1B)

as well as the legend:

(B) hnRNPK co-immunoprecipitates with ectopic wt Fbxo4 and ΔF Fbxo4- catalytically deficient mutant unable to recruit E2 ligase but capable of substrate recognition

Its function as a negative control for ubiquitylation has been also recalled in the line 116 where ubiquitylation data is described:

... SCF^{Fbxo4ΔF} (deletion of F-box inhibits recruitment of core E3 ligase components) catalyzed hnRNPK...

- Fig S1C: Please, spell-out the domains of both proteins and explain their roles in the legend. Domain name has been extended in the legend as follows:

(C) Schematic representation of hnRNPK: K-homology domains 1 – 3; KH1, KH2, KH3), K interacting domain (KI) and K nuclear shuttling (KNC) nuclear localization signal (NLS) and Fbxo4: Fbxo4 dimerization domain (D-domain), F-Box cassette, linker region, substrate binding domain) domain structures with indication of mutants utilized to determined protein recognition regions

- Lane 116: I guess the authors refer to Fig S2B (not S1B) Indeed, S2B was meant to be referred. Error has been corrected. (lane 117)
- Fig S2C: Please show more contrasted panels for hnRNPK. The protein is barely visible in several cell lines. We provided higher quality data New figure:

old version

- Figure S2F: There is almost no difference in the ubiquitylation of hnRNPK between Fbxo WT and delta-F. Please, quantify, correct for input hnRNPK and indicate at bottom of WB. Western blot has been quantified and values displayed below the Myc blot.

- Figure S2H: Explanation missing in Figure legend.
Fig S2J (S2H before revision) has been described in the legend.

(J) The effect of SCF^{Fbxo4} on hnRNPk cytoplasmic/nuclear distribution was evaluated by fractionation of HEK293T cells; wt of K21/34R myc-hnRNPk mutant was co-overexpressed with either wt or catalytically inactive ΔF Flag-Fbxo4.

- Lane 147: S3C (not SC3)
Error has been corrected

- Figure S2I is not mentioned in the main text
Fig S2K (before revision S2I) is mentioned in the discussion section, lane 354.

- Figure S3D-E: I don't see much of a rescue of Myc expression by over-expression of ubiquitin-refractory hnRNPk, as mentioned in the main text. Please, quantify, correct for hnRNPk expression (myc-K), and indicate at bottom of WB.
The western blot has been quantified and values displayed below the c-Myc blot. Obtained values have been corrected to myc-K western blot and to loading control (Vinculin).

- Figure 3E: Scatter plots for hnRNPk KD section are switched.
The position of scatter-plots has been confirmed to be correct, according to applied criteria knock-down of hnRNPk in wt cells was expected to induce changes opposite to Fbxo4 knock-out.

- Lane 189: Figure 3K addresses the role of Fbxo4 in promoting the synthesis of endogenous c-Myc, but says nothing about hnRNP. Please, eliminate 'hnRNP' from the sentence.

Indicated sentence has been corrected as suggested (lane196).

- Lane 204: I guess the authors refer to Fig S3M (not Fig 3).

Indeed, Fig S3M was meant to be indicated. Text has been corrected.

- Lane 2013: Fig 4A,B represents the analysis of the 700 Fbxo4-hnRNP targets? If so, please indicate clearly by adding the number, as right now one could think that any target deemed as dysregulated in Fig 3E counts. If not, please show the analysis of the 700 targets.

The 700 genes identified in 3E were taken as an input for pathway enrichment analysis. We modified indicated sentence to clearly state that information.

New sentence (lane227):

To predict potential phenotypic output of SCF^{Fbxo4}-hnRNP differentially regulated genes, all 693 putative targets identified in Fig. 3E were used as an input in the pathway enrichment analyses using KEGG, Reactome and gene ontology (GO).

- What is the status of endogenous Fbxo4 in B16F10 cells? Is it wt or mutated?

Sanger sequencing of Fbxo4 cDNA derived from B16F10 RNA indicates on wild type status of the gene.

Mus musculus F-box protein 4 (Fbxo4), mRNA

Sequence ID: NM_134099.2 Length: 3479 Number of Matches: 1

Range 1: 150 to 1307 GenBank Graphics Next Match Previous M

Score	Expect	Identities	Gaps	Strand
2139 bits(1158)	0.0	1158/1158(100%)	0/1158(0%)	Plus/Plus
Query 1	ATGGCTGGAAGCGAGCCCGCGGAGCCGGCTCCCGCCGCCCGCCAGCGACTGGGGCCGC	60		
Sbjct 150	ATGGCTGGAAGCGAGCCCGCGGAGCCGGCTCCCGCCGCCCGCCAGCGACTGGGGCCGC	209		
Query 61	CTGGAGGCAGCCATCTGAGCGGCTGGAGGACCTTCTGGTATTCGGTGGCCAAGGAGCGG	120		
Sbjct 210	CTGGAGGCAGCCATCTGAGCGGCTGGAGGACCTTCTGGTATTCGGTGGCCAAGGAGCGG	269		
Query 121	GCGACGCCGACGGCTCCCGAAGGAGGCGGCGGAGGAGACGAGCGCGCTGACGCGGCTG	180		
Sbjct 270	GCGACGCCGACGGCTCCCGAAGGAGGCGGCGGAGGAGACGAGCGCGCTGACGCGGCTG	329		
Query 181	CCGGTTGATGTGCACTGTATATCTTGTCTTTCACCCACGATCTGTGCCAGCTG	240		
Sbjct 330	CCGGTTGATGTGCACTGTATATCTTGTCTTTCACCCACGATCTGTGCCAGCTG	389		
Query 241	GGAGTACAGATCATTACTGGAACAAAATAAGAGACCAATTCTCTGGAGATACTTT	300		
Sbjct 390	GGAGTACAGATCATTACTGGAACAAAATAAGAGACCAATTCTCTGGAGATACTTT	449		
Query 301	CTGTGGGGATCTCCCTTCTGGTCTCGGTTGATTGGAAGTCACTTCCAGATCTAGAG	360		
Sbjct 450	CTGTGGGGATCTCCCTTCTGGTCTCGGTTGATTGGAAGTCACTTCCAGATCTAGAG	509		
Query 361	ATCTTAAAAAGCCAATATCTGAGGTCACCGACAGCACTTGTCTTGATTACATGGAGGTT	420		
Sbjct 510	ATCTTAAAAAGCCAATATCTGAGGTCACCGACAGCACTTGTCTTGATTACATGGAGGTT	569		
Query 421	TATAAATGTGCTGCCATATACGCGAAGAGCCTTGAAAGCCAGCCGCTCATGTATGGA	480		
Sbjct 570	TATAAATGTGCTGCCATATACGCGAAGAGCCTTGAAAGCCAGCCGCTCATGTATGGA	629		
Query 481	GTGGTTACCTCTTTCTTACACTCACTGATCATTGAGATGAAACCCGGTTTGTATGTTT	540		
Sbjct 630	GTGGTTACCTCTTTCTTACACTCACTGATCATTGAGATGAAACCCGGTTTGTATGTTT	689		
Query 541	GGACCAGGTTTGAAGAAGTGAACACATCCTTGGTGTGAGTTGATGCTCTTGAGGAC	600		
Sbjct 690	GGACCAGGTTTGAAGAAGTGAACACATCCTTGGTGTGAGTTGATGCTCTTGAGGAC	749		
Query 601	CTTTGCCAACTGCTGGTTTACCTCACAGACAGATTGATGGTATTGGATCTGGAGTCAAC	660		
Sbjct 750	CTTTGCCAACTGCTGGTTTACCTCACAGACAGATTGATGGTATTGGATCTGGAGTCAAC	809		
Query 661	TTCCAGTTGAATAACCAGCAAAAATCAACATCCTGATATTACTCGACTACCAGAAAA	720		
Sbjct 810	TTCCAGTTGAATAACCAGCAAAAATCAACATCCTGATATTACTCGACTACCAGAAAA	869		
Query 721	GAAAGAGACAGAGCAAGGGAGGAGCACACCAGCACCGTTAAACAAGATGTTAGCCTACAG	780		
Sbjct 870	GAAAGAGACAGAGCAAGGGAGGAGCACACCAGCACCGTTAAACAAGATGTTAGCCTACAG	929		
Query 781	AGTGAGGGGACGAGCAGCAGGGCAGCCGCTACAGTGTGATCCCGCAGATTGAGAAAGTG	840		
Sbjct 930	AGTGAGGGGACGAGCAGCAGGGCAGCCGCTACAGTGTGATCCCGCAGATTGAGAAAGTG	989		
Query 841	TGTGAAGTCGTAGACGGGTTTCTACGTGGCAACCGCTGAAAGCTCACCAGCTCATGAA	900		
Sbjct 990	TGTGAAGTCGTAGACGGGTTTCTACGTGGCAACCGCTGAAAGCTCACCAGCTCATGAA	1049		
Query 901	TGGCAAGATGAATTTCTCGGATTATGGCCATGACAGACCAGCTTTTGGATCTTCAGGA	960		
Sbjct 1050	TGGCAAGATGAATTTCTCGGATTATGGCCATGACAGACCAGCTTTTGGATCTTCAGGA	1109		
Query 961	AGACCCATGCTGGTTTTATCTTGTATTTCTCAAGCAGATGAAAGAGAATGCCCTGTGTTT	1020		
Sbjct 1110	AGACCCATGCTGGTTTTATCTTGTATTTCTCAAGCAGATGAAAGAGAATGCCCTGTGTTT	1169		
Query 1021	TATTTAGCTCATGAGCTGCACCTCAGTCTTCTAAACCACCATGGATGGTCCAGGATACA	1080		
Sbjct 1170	TATTTAGCTCATGAGCTGCACCTCAGTCTTCTAAACCACCATGGATGGTCCAGGATACA	1229		
Query 1081	GAGGCTGAAACTCTGACTGGTTTTTTGAATGGCATTGAGTGGATTCTTGAAGAAGTAGAA	1140		
Sbjct 1230	GAGGCTGAAACTCTGACTGGTTTTTTGAATGGCATTGAGTGGATTCTTGAAGAAGTAGAA	1289		
Query 1141	TCTAAGCGTGCAAAATGA 1158			
Sbjct 1290	TCTAAGCGTGCAAAATGA 1307			

• Lane 287: I guess the authors refer to translational control (not post-translational). Inherently both are right, since lack of transcriptional dysregulation suggests potential dysregulation through protein synthesis or by posttranslational modifications. In the sentence we expressed posttranslational modification since our work proposes the model of ubiquitylation. We agree that translational control is a vital option that should be mentioned therefore we adjusted the sentence as follow (lane 300):

In contrast, no significant changes in hnRNPk expression are apparent emphasizing the importance of translational or posttranslational control over expression level (Fig S6G).

- Lane 294: I guess the authors refer to Fig 6F (not 6G).

Reviewer suggestion is correct, Fig 6F was meant to be indicated. Text has been updated accordingly (lane 309).

- Figure 6H: What is the difference between the right and left panels?

Both graphs refer to data submitted by two different authors to Oncomine database, the name of reference has been added above the graphs.

Reviewer #2 (Remarks to the Author):

In this manuscript, Mucha et al. describe a role of RBP hnRNPk in regulation of translation, in particular of the proto-oncogene c-MYC, which promotes hnRNPk function in tumorigenesis. The authors suggest that the oncogenic function of hnRNPk depends on its cytoplasmic function that is regulated by ubiquitylation. Ubiquitylation is proposed to limit the ability of hnRNPk to bind to its target mRNA including c-MYC RNA.

My review is focused on the experiments that establish the E3 ligase-substrate relation between hnRNPk and SCF-Fbxo4. The authors show relatively convincingly that hnRNPk can be ubiquitylated by Fbxo4 in vitro, however the in vivo evidence remains scarce and not convincing. Also, that hnRNPk is modified by K63-linked ubiquitylation is not convincing. In general, I have problems with the quality of the western blots that demonstrate the interaction between hnRNPk and Fbxo4 as well as ubiquitylation of hnRNPk in vitro and in vivo (especially the ones shown in extended data).

Figure 1H demonstrates the interaction between hnRNPk and Fbxo4. I appreciate that this is an IP with endogenously expressed proteins, however the band with hnRNPk is hardly visible. This IP should be done in more replicates and preferably with quantification and also blotting for a positive control such as CCND1.

Higher quality picture from different experiment has been provided. A published previously target of SCF-Fbxo4 – Fxr1 (1) is used as a positive control.

Since endogenous co-IP usually is compared to negative background control such as non-specific IgG we found quantification of this experiment to be not relevant as the best way of western blot quantification is comparison to reference band which by principle of endogenous co-IP is not available.

1- 1- Qie, Shuo, et al. "Fbxo4-mediated degradation of Fxr1 suppresses tumorigenesis in head and neck squamous cell carcinoma." *Nature communications* 8.1 (2017): 1-14.

Two panels Fig. 2F and G show evidence for in vivo ubiquitylation of hnRNPk, however the ubiquitylation pattern looks completely different. I could not find any information about the cell lines used for these assays and how the distribution of endogenous hnRNPk looks in these cell model and whether this localization pattern is resembled when using tagged hnRNPk. This is essential if the authors want to make claims about the cytoplasmic vs nuclear function and use both endogenous and overexpressed hnRNPk.

The difference in the pattern between Fig 2F and G is due to different set up. In the ubiquitylation assay presented in Fig 2F, cells were lysed in our optimized ubiquitin assay lysis buffer followed by

denaturation. In contrast, samples in experiment Fig 2G were first fractionated utilizing milder lysis buffers and included differential washing steps. In Fig S2J we provide the distribution of both myc-tagged ectopic and endogenous hnRNPK is the same and does not change upon Fbxo4 overexpression; to connect those two aspects an additional reference in the results section has been made (bolded), Lane 143:

Therefore, we addressed whether Fbxo4-dependent ubiquitylation may define hnRNPK trafficking. Fractionation of cells showed no changes in hnRNPK localization for either wild type or ubiquitin-refractory hnRNPK mutant (Fig S2J). In vivo ubiquitylation assay showed presence of ubiquitylated hnRNPK only in cytoplasm (Fig 2H).

The authors claim that hnRNPK is modified by K63-linked chains (which is atypical for Fbxo4) by using ubiquitin mutants but never test whether ubiquitylation of hnRNPK is affected upon proteasome inhibition. I would expect that it is not if these are indeed K63-linked chains. As it is now, these panels are meaningful without knocking down hnRNPK and showing that these smears are really on hnRNPK (especially since two completely different patterns of ubiquitylation are observed).

Can the authors try K48 vs K63 Ub-specific antibodies?

We appreciate the suggestion to block proteasome which is standard in classic ubiquitylation studies, although as we noticed lack of changes in the steady state level of hnRNPK upon Fbxo4 knock-down (Fig. S2C) and no significant changes in hnRNPK stability between wt and Fbxo4 null MEFs (Fig S2D) While the Ub proteasome system is usually prominent in proteins with the fast turnover like c-myc or cyclin D1, the half-life of hnRNPK reaches over 15 hours. Together, these observations prompted us to focus on non-proteasomal regulation of hnRNPK. Lastly, we found no increase in binding between endogenous hnRNPK and Fbxo4 upon MG132 treatment which data has been added to the manuscript as a Fig S2E:

Lane 121: *Proteasome inhibition did not increase binding of endogenous hnRNPK and Fbxo4 as would be expected in case of degradation control (Fig S2E).*

The difference in pattern between in vivo and in vitro ubiquitylation experiments was observed in previous studies on SCF-Fbxo4 substrates (1) and is common in ubiquitylation studies. In in vitro assays (Fig 2B, C, E, S2A, S2F), as we elaborated on in the previous comment, we purify all substrates and reconstitute the reaction in energy buffer then blot for the substrate(hnRNPK) looking for gradual increase of molecular weight. For in vivo assays we overexpress myc-tagged hnRNPK and perform immunoprecipitation using myc-tag specific antibodies. To make sure the smearing is specific to poly-ubiquitin-hnRNPK we present a spectrum of controls in Fig 2A including inactive Fbxo4 overexpression (ΔF), inhibitor of cullin activation (MLN9424) and empty vector (background control). In addition, we boil lysate in 1% SDS prior to IP in order to disrupt all non-covalent binding to avoid detection of ubiquitylated binding partners co-immunoprecipitated with hnRNPK rather than hnRNPK itself. Such a strict condition eliminates the use of the most common K63 specific

antibody HWA4C4 which is temperature sensitive. We were able to detect K63 ubiquitin chains with D7A11 K63-specific Cell Signaling antibody (#5621) but the quality made the data inconclusive.

1- Qie, Shuo, et al. "Fbxo4-mediated degradation of Fxr1 suppresses tumorigenesis in head and neck squamous cell carcinoma." Nature communications 8.1 (2017): 1-14.

Fig 2C shows in vitro ub of hnRNPK by either K48 or K63-linked chains. I find it surprising that the chain pattern look exactly the same on the WB except that K48 is less abundant. This would suggest that the authors look at the same chain type but in one case hnRNPK is less efficiently ubiquitylated. Since analysis of ubiquitin was performed by blotting of the substrate and resolved on denaturing SDS-PAGE gel followed by blotting for the substrate not ubiquitin, we expect to see upward laddering due to increase of molecular weight rather than a changed pattern that is characteristic for antibodies recognizing ubiquitin linkages under non-denaturing conditions. The in vitro ubiquitylation assay does not include immunoprecipitation step because we used purified components, therefore blotting with ubiquitin antibody generates background signals from E1-Ub conjugates which impairs proper detection of substrate-poly ubiquitin conjugates. We included new data from in vitro ubiquitylation assay using KO ubiquitin (all lysines mutated to arginine) as a background control. Here we present the signal from reaction with K48 only mutant is at similar level to KO while K63 only is significantly higher.

To strengthen our data, we performed an in vivo ubiquitylation assay in HEK293T cells including the same mutants (KO, K63 only and K48 only). Again, the signal level from K48 only mutant is similar to background control KO while having available K63 only lysine, SCF-Fbxo4 dependent ubiquitylation of hnRNPK is strikingly high.

Extended data on hnRNPK ubiquitylation are very confusing and most of the WB in extended Figure 1 and 2 do not seem to convincingly support the claims.

For example:

Fig.S1A – Fbxo4 does not seem to be enriched in hnRNPK IP.

Fig S1A has been replaced to present higher quality.

new version:

Old version

Fig. S1B – hnRNPK is enriched in WT and deltaF Fbxo4 and this is not the case in main figure. There also seems to be a strong double band in the delta F mutant that resembles the size of the WT protein.

Delta F mutant is catalytically deficient Fbxo4 due to loss of ability to recruit E2 ligase (deficient in Skp1 binding as previously published (1)) but maintains the ability to recognize substrate through its C-terminus, which is consistent with all previously identified SCF-Fbxo4 substrates (2 - 4). In the main figures delta F is widely utilized as a negative control that can bind substrate without catalytic activity. In Fig S1B, a shorter exposure of the Flag-Fbxo4 western blot has been presented to clearly show that the non-specific band is higher than ectopic wt Flag-Fbxo4.

New version

old version

1- Schulman, Brenda A., et al. "Insights into SCF ubiquitin ligases from the structure of the Skp1–Skp2 complex." *Nature* 408.6810 (2000): 381-386.

2- Lee, Tae Ho, et al. "The F-box protein FBX4 targets PIN2/TRF1 for ubiquitin-mediated degradation and regulates telomere maintenance." *Journal of Biological Chemistry* 281.2 (2006): 759-768.

3- Qie, Shuo, et al. "Fbxo4-mediated degradation of Fxr1 suppresses tumorigenesis in head and neck squamous cell carcinoma." *Nature communications* 8.1 (2017): 1-14.

4- Barbash, Olena, et al. "Mutations in Fbx4 inhibit dimerization of the SCFFbx4 ligase and contribute to cyclin D1 overexpression in human cancer." *Cancer cell* 14.1 (2008): 68-78.

Fig. S1D – Fbxo4 is barely seen in the WT lane and seems that none of the mutants bind to hnRNP K. What is the explanation for that if only one terminus is important for binding?

Loss of binding between hnRNP K and truncated C-termini Fbxo4 is expected as that is established substrate recognition region for all known Fbxo4 targets and majority of F-box proteins. Loss of binding in case of N-terminal deletion suggest efficient binding requires active homodimer form of Fbxo4 which is mediated through N-terminal dimerization-domain.

Higher quality data has been provided with additional background controls (lane1):

Fig. S1E – Again the WT band is barely visible and the delta N mutant binds stronger than the WT. Is there any explanation for this?

Fbxo4 is exclusively localized to the cytosol while hnRNP K is predominantly nuclear with relatively low fraction maintained in the cytosol. As presented in the Fig S1C cartoon, N-terminal deletion of hnRNP K removes the nuclear localization signal from hnRNP K that leads to cytosolic accumulation and an increase in binding of hnRNP K to Fbxo4.

Contrast of Fbxo4 western blot has been adjusted to improve visibility:

New version:

Old vesion:

In Figure S2 the authors show the dependency of ubiquitylation on K21 and K34 in hnRNPk. However, in panel E the ubiquitylation is completely gone in single K34R mutant and not really affected by K21R mutant. This seems to be then opposite in panel F where K34R barely affects ubiquitylation.

Indeed, we observed different patterns of ubiquitylation loss when comparing data from the in vivo and in vitro assays with regards to K21 and K34. We do not have clear explanation for this observation but to provide additional data to validate the proposed ubiquitylated residues, we removed the entire region between lysine 21 and 34 and observed complete loss of hnRNPk ubiquitylation using an in vitro ubiquitylation assay. New data has been included as a Fig S2H.

And described in the result section as followed (lane 132):

While in vitro ubiquitylation did not reveal a robust loss of signal in the K34R mutant (Fig S2G), truncation of the hnRNPk N-terminal region (Fig S1C) resulted in complete lack of ubiquitylation signal (Fig S2H) consistent with our interpretation that the differential pattern of signal loss reflects differences in the nature of the in vivo vs. in vitro experiments

None of the CHX chase assays (either for MYC or hnRNPk) are accompanied by quantification or determination of half-lives. This is also true for all other WB. This should be at very least provided for key data where the representative WBs do not show a clear result.

CHX chase assays have been quantified in reference to the 0 time point. Quantification has been added to Fig, S2D, S3H (according to the reviewer's request) and additionally to 3C,S3D, S3E S5C

S3H

S2D

S5C:

3C:

S3D, E

E

The authors use the previously published MS interactome to show that hnRNPK is a putative substrate of Fbxo7 and this is only indicated in Methods section. I find this misleading. Also, no information about the number of replicates and statistical analysis is provided. Simply providing a table with selected proteins and number of identified peptides is not acceptable. I suggest that the authors completely remove the previously published interactome analysis from the figure and simply mention that hnRNPK was identified as putative substrate in a previous study with indicated reference.

Mass-spectrometry was performed from single samples of material from immunoprecipitation of ectopic wt Fbxo4 or ectopic Fbxo4 Δ F expressed in Fbxo4 null MEFs treated with DMSO or proteasome inhibitor (MG132). The previous article presents analysis of single set of excised bands that revealed Fxr1 as a Fbxo4 target. Here we present extended analysis of the remaining material from the original experiment and present novel targets which have not been published. To eliminate any confusion and to be fully transparent we followed reviewer's suggestion and removed the mass spec procedure from the methods section and replaced it by the quote of previous work where samples were collected. In the result section the sentence:

We also assessed binding of RBPs to Fbxo4 via its purification followed by mass spectrometry analysis. Multiple KH-domain RBPs co-purified with Fbxo4 with the highest peptide coverage for hnRNPK (Fig 1E).

was replaced by (lane 93):

Analyzing material collected during a previous purification/mass spectrometry experiment in which Fxr1 was identified as a Fbxo4 target (21), we noted multiple KH-domain RBPs that co-purified with Fbxo4 with hnRNPK as one of high potential relevance (Fig 1E).

MLN4924 is misspelled in Fig. 1A.

Spelling has been corrected

The figure legends are in general too briefly explained without providing any details on number of replicates and statistics.

Replicates numbers and statistical test information has been added to Fig. 3, S3, 4, 5, S5, 6, S6.

Fig 3

The data in (I, M) represents mean \pm SD and was analyzed by two-tailed Student's t-test (n=3). The data in (J) represents mean ratio (control/Fbxo4) and was compared by ratio paired Student's t-test (n=4). In (E) all selected targets $MEF^{Fbxo4^{-/-}}$ vs. MEF^{wt} complied with adj. p-Value <0.05 for either RNA and RPF, the rescue comparisons were included based on the fold change. Data presented in the A,D,E,F,G is based on the RNA-seq and Ribo-seq data run in biological duplicates. Correlation between replicates and PCA analysis of RNA and Ribo-seq is presented in Fig S8. Detailed list of targets, fold change and rescue levels presented in (E) is available in supplementary material 1.

Fig S3

*Data in C, K, M, N represents mean \pm SD and was analyzed by Student's t-test (M – n=4, N – n=3) or one way ANOVA (C – n=4). Data presented in J (n=3) shows mean ratio (control/K siRNA sample) and was compared by paired ratio Student's t-test. *p<0.05, **p<0.01, ****p<0.0001. In (M) and (N) last time point was subjected to statistical comparison.*

Fog 4

The data from invasion experiments represents mean \pm SD and was analyzed by two-tailed Student's t-test (D - n=4, F - n= 3). In cell motility assay 149-150 cells were tracked per condition in three independent experiments (~50 cells/experiment). Data represents median with 95% CI and were compared by two-tailed Student's t-test. *p<0.05, **p<0.01, ***p<0.001, ****p<0.0001

Fig 5

All data represents mean \pm SD and was analyzed by Two-way ANOVA (C, H; n= 10 and 4 respectively) or Student's t-test (F, G, I, K), *p<0.05, **p<0.01. Scale bars: black – 100 μ m, red -20 μ m. Analysis of F and J was done on n=3 for empty vector, n=3 for Fbxo4 OE and n=4 for K RNAi. Analysis of J and K was run on n=4 for vehicle and JQ1 treated group. C-Myc quantification was done on 3 representative picture (40x) from each staining using ImageJ and reciprocal intensity approach (maximum intensity-measured intensity; a.u - arbitrary unit).

Fig S5

RNA extracted from WM983B melanoma with Fbxo4 knock-down and control cells were subjected to RNAseq (n=3) which revealed ~5000 differentially regulated genes (E-I) subsequently used for functional modeling by Reactome, KEGG, Gene Ontology: Molecular Function (GO_MF), Cellular Compartment (GO_CC), Biological Process (GO_BP) resource. *- corrected p-value<0.05 (padj).

Fig6

In IHC quantification of all TMAs the 0 -3 scale was applied (0-negative staining, 1-low, 2-med, 3-hig) results summarized in violin plots presenting distribution of score and analyzed statistically by non-parametric Mann-Whitney test. Immunofluorescence is presented as a mean fluorescence intensity \pm SD from 200 randomly selected cells/individual and statistically compared by two-tailed Student's t-test, *p<0.05, **p<0.01, ***p<0.001, ****p<0.0001. Scale bars – black 50 μ m; red- 200 μ m.

Fig S6

(E) The cancer proteome atlas data (tcpportal.org) shows lower survival probability for group of patients with high c-Myc protein level (Log-Rang P < 0.05). (F) Images of all tumors extracted in B16F10 melanoma allograft model with JQ1 inhibitor treatment. (G) GTEx and TCGA data shows no significant difference in expression of hnRNPK between melanoma patients and normal tissue. (H)

There is no legend for figure S2H.

Fig S2J (S2H before revision) has been described in the legend

(I) Analysis of ectopic and endogenous hnRNPK distribution between nucleus and cytoplasm in HEK293T cells expressing wt or Δ F Fbxo4

Reviewer #3 (Remarks to the Author):

In this manuscript, Diehl and co-workers report a molecular mechanism by which the ubiquitin pathway restricts oncogenic activity of hnRNPK. They identified hnRNPK as a target of the ubiquitin ligase SCF-Fbxo4, which ubiquitylates hnRNPK through K63 linkages rather than the degradative K48 linkages. By using a genome-wide screening, they identified c-Myc as a target of SCF-Fbxo4-hnRNPK. Loss of Fbxo4 triggers hnRNPK-dependent increasing in c-Myc translation and tumorigenesis. Taken together, the work by Mucha et al. demonstrated an essential role of the ubiquitin ligase Fbxo4 in limiting hnRNPK function to maintain appropriate translational homeostasis.

Overall, this is a well-written manuscript, demonstrating an important role for a ubiquitin ligase in c-Myc translational regulation and limiting tumorigenesis. The genome-wide screenings and analyses are convincing, and the functional significance of the Fbxo4-hnRNPK in vivo and in human subjects is clearly shown. I have only a few concerns regarding the molecular mechanism of this regulatory axis, as described below.

1) The authors tried to demonstrate in vitro that SCF-Fbxo4 ubiquitylates hnRNPK through K63-linkages. However, the ubiquitin chain specificity generated by the RING-type E3s depend on the property of the E2 enzyme. As UBC5 is a promiscuous E2, it can assemble ubiquitin chains either through K48, K63, or K11 linkages. More convincing evidence in cell is needed.

That is interesting question, nonetheless the primary goal of this investigation was to identify RNA binding proteins controlled by SCF-Fbxo4. Exploring E2 ligases in this context is tempting but, in our opinion, reaches beyond the scope of this study. Here we focus on E3 providing substrate specificity. As the reviewer emphasized, E2 was previously presented to cooperate with the Skip1-Cul1-Rbx1 complex which supports our conclusion of having the ability to catalyze K63 linkages assembly.

2) It is unclear from the manuscript how the K63-linked ubiquitylation affects the function of hnRNPK.

We postulate that K-63 linked ubiquitylation suppresses translation of c-Myc. That hypothesis is based on the following crucial observation:

- ubiquitylation attenuates affinity hnRNPK to RNA (Fig. S3L)
- ubiquitin deficient mutant is able to activate translation from 5'UTR region of c-myc (Fig3I-J)
- ubiquitin hnRNPK mutant binds relatively more c-Myc mRNA than wt hnRNPK (Fig. 3M)
- loss of hnRNPK decreases c-Myc protein simultaneously increasing amount of ribosome protected fragment at specific region (Fig 3G).

We note the lack of data that would directly show the impact of hnRNPK ubiquitylation on c-Myc translation. To demonstrate the relation clearly, we performed an additional experiment showing Fbxo4 dependent ubiquitylation of hnRNPK inhibits translation of c-Myc. We designed the experiment in HEK293T cells using overexpressed Fbxo4 to enforce hnRNPK ubiquitylation or the catalytically inactive mutant Fbxo4 Δ F as a negative control. Cell lysates were used for two independent immunoprecipitations (1 set - c-Myc, 2 set – myc-hnRNPK) to simultaneously monitor c-Myc synthesis rate and hnRNPK ubiquitination status. Ubiquitylation was carried out according to our standard in vivo Ub assay procedure (IP myc-hnRNPK with 9B11 myc- tag specific antibody and immunoblot for HA-ubiquitin). Tracking of c-Myc synthesis was performed by click chemistry in two step-procedure: 1) 3-time point chase of methionine analog AHA (L-azidohomoalanine) incorporation followed by c-Myc immunoprecipitation (IP with antibody specific to endogenous c-Myc oncoprotein); 2) chemoselective ligation (click reaction) between AHA and alkyne-tetramethylrhodamine (TAMRA). Detection of *de novo* synthesized c-Myc was performed by SDS-

PAGE with subsequent immunoblotting with an anti-TAMRA antibody. A cartoon of the experimental workflow is presented below along with the data that demonstrate ubiquitylated hnRNPK correlates with limited c-Myc synthesis compared to negative control.

We believe the above data show a clear correlation between SCF-Fbxo4 dependent ubiquitylation of hnRNPK and translation of c-Myc. Finally, we wrote an additional paragraph in the results section of the manuscript to summarize our observation and deliver mechanistic insight in the work (lane 209):

Overall, the biochemical role of hnRNPK ubiquitylation can be defined by the following crucial observations: 1) ubiquitylation lowers hnRNPK's affinity to RNA (Fig S3L) including c-Myc mRNA (Fig 3M); 2) ubiquitin hnRNPK mutant stimulates c-Myc translation through 5'UTR IRES (Fig 3I-J); 3) ubiquitylation status of hnRNPK correlates with a lower c-Myc synthesis rate (Fig 3L); altogether these observations demonstrate that according to the expected tumor suppressor nature of Fbxo4, SCFFbxo4 functions as a limiting factor for hnRNPK oncogenic translation.

3) In Fig. 3B and Fig. S3D, a portion of Fbxo4 remains in the Fbxo4 ^{-/-} cells, while hnRNPK is completely lost in the knockdown cells. Please explain this. I guess they might be mislabeled? The observed band is non-specific which unfortunately runs right above endogenous Fbxo4. In both cases we re-sized the Fbxo4 western blot and added additional arrows with an asterisk to distinguish between Fbxo4 and the non-specific signal.

Fig 3B

Fig S3D

4) In Fig. 2G, the anti-HA blot is of poor quality. I suggest that anti-ubiquitin blot of the same IP may improve the quality.

In Fig2G, Extraction of nuclear and cytoplasmic fractions requires multiple steps and is performed in buffers that are not optimal for standard in vivo ubiquitylation assays. In the process of optimizing this assay, we found the HA antibody provides a cleaner signal over FK2 or P4D1 ubiquitin antibodies. To improve quality of Fig 2G we provided better exposure with additional contrast adjustment.

G

Minor comment

Fig. 3K: Mg132 should be corrected to MG132.

The error has been corrected

REVIEWERS' COMMENTS

Reviewer #1 (Remarks to the Author):

The manuscript by Mucha et al has improved with the new additions. The data report an important connection between Fbxo4-mediated ubiquitylation of hnRNPk, c-Myc expression and tumorigenesis. However, I remain unconvinced about the molecular mechanisms regarding ribosomal pausing. The text could be adjusted to tune-down this conclusion. Specific comments regarding prior concerns follow:

Main comments:

1) Figure 3G: The ribosome profiling data claiming pausing at one specific site in the ORF is still very weak. As sequence depth is not sufficient to reveal codon-by-codon movement of the ribosome, a translation-dependent stalling of the ribosome at this position cannot be concluded. Given variations in RPF accumulation at other positions of the ORF (e.g. positions 837-913 in the top graphic or position 1297 in the bottom one) and the fact that only 2 replicates have been performed per condition, variations at the proposed ribosome stalling site could have arisen just by chance. Furthermore, the high amount of ribosomes detected in the 3' UTR (new Fig S9), poses the question of how many of the RPFs in this study really correspond to translating ribosomes. I understand that the authors would prefer to keep this panel in order to reconcile the marked ribosome occupancy of the c-Myc transcript with the reduction in protein production, but the current data do not allow any solid claim on translational stalling. Therefore, I suggest to tune-down the ribosome pausing hypothesis, moving Figs 3F-G to supplementary data and acknowledging that these are just preliminary data. Please, also remove 'ribosome pausing' from the heading in lane 155 and modify the reading of this section accordingly. (By the way, I failed to see the precise nucleotide number of the 'stalled' peak indicated in the figure).

On another note, as the authors recognize, regulation of IRES-mediated translation is independent to ribosomal stalling. Thereby the statements in lanes 191-192 '...suggests that SCFFbxo4-dependent regulation of hnRNPk activity contributes to pausing at these sites. To address the mechanism, we considered that SCFFbxo4 might regulate hnRNPk-dependent c-Myc protein synthesis at its Internal Ribosome Entry Site (IRES)' are confusing. Re-writing this section to tune-down the ribosomal stalling part should solve this.

2) Table S1: The Table contains 439 genes, of which 53 are Myc targets. These numbers do not fit with the main text, where approx 700 genes are mentioned, of which 77 are Myc targets. Please, add the remaining genes to the Table or correct your statement.

3) Figure S3L: The new Figure is much better, thank you. As a frequent user of PNK and CLIP methods, though, I usually see that high RNase concentrations lead to compression of the 32P-RNA smear above the size of the studied protein into a sharp band at the size of the studied protein, which is surprisingly absent in this figure. Probably the RNA is over-digested, but to really ensure that hnRNPk is present, and that the band indicated by the arrow actually corresponds to hnRNPk, the very same 32P-RNA membranes should be subjected to Western blot against myc-hnRNPk. The band of hnRNPk should exactly coincide with the arrow, and should also be present in the H lanes.

Minor comments:

Lane 101: 'We noted that the RNA binding motifs for hnRNPk were highly enriched in mRNAs dysregulated upon Fbxo4 ablation (Fig 1F)'. It is more correct to say 'We noted that mRNAs dysregulated upon Fbxo4 ablation were highly enriched in hnRNPk motifs'.

Figure 1C: Please, mention in the main text that oRNAment analysis on targets responding at the level of RPF has also been run and show the same (3rd) position of hnRNPk in terms of motif abundance within KH domain RBPs (data not shown).

Figure S1D: Please, mention in the figure legend or in main text that N-terminal truncations are expected to result in impaired substrate recognition because functional/active SCF-Fbxo4 requires dimerization mediated by the N-terminal D-Domain.

Figure S5A: Can the authors, please, show quantification of several experiments? This would strengthen a reduction in hnRNP-K-Fbxo4 interaction upon Fbxo4 mutation.

Reviewer #2 (Remarks to the Author):

The authors satisfactorily addressed all my concerns.

Reviewer #3 (Remarks to the Author):

In the revised manuscript, the authors have addressed most of the previous concerns. No further question.

Reviewer #1 (Remarks to the Author):

The manuscript by Mucha et al has improved with the new additions. The data report an important connection between Fbxo4-mediated ubiquitylation of hnRNPk, c-Myc expression and tumorigenesis. However, I remain unconvinced about the molecular mechanisms regarding ribosomal pausing. The text could be adjusted to tune-down this conclusion. Specific comments regarding prior concerns follow:

We agree with the reviewer that phrasing in the previous manuscript version suggested more certainty about the underlying mechanism than warranted by the data. Accordingly, we have re-phrased this section, as rationalized in detail in our response to the specific points.

Main comments:

1) Figure 3G: The ribosome profiling data claiming pausing at one specific site in the ORF is still very weak. As sequence depth is not sufficient to reveal codon-by-codon movement of the ribosome, a translation-dependent stalling of the ribosome at this position cannot be concluded. Given variations in RPF accumulation at other positions of the ORF (eg. positions 837-913 in the top graphic or position 1297 in the bottom one) and the fact that only 2 replicates have been performed per condition, variations at the proposed ribosome stalling site could have arisen just by chance. Furthermore, there high concentration of ribosomes detected in the 3' UTR (new Fig S9), poses the question of how many of the RPFs in this study really correspond to translating ribosomes.

The reviewer is correct; the data do not reveal codon-level occupancy by the ribosome. With that in mind, translation-dependent stalling of the ribosome at this position cannot be concluded. We note, however, that we did not claim this in the manuscript. We described the RPF accumulation as ribosome pausing. We realize that this term might be misinterpreted as translation-dependent stalling of the ribosome at this position, and in order to avoid potential confusion, we have removed this term. We are now careful terming the increased RPF density ribosome accumulation, which it clearly is. While it is possible that the increased ribosome accumulation is caused by translation-dependent stalling of the ribosome at this position, or by stalling downstream from this position, we have now clearly marked this as a *possible* interpretation of the data.

The marked ribosome occupancy in the 3'UTR is frequently seen in ribosome profiling experiments conducted with MNase. However, it has been shown in numerous systems that ribosome occupancy in the ORF does not markedly differ between different nuclease treatments, suggesting that the 3'UTR occupancy is, in a not well understood manner, related to ORF translation, not to random ribosome binding to non-translating transcripts. Nevertheless, we (and most other ribosome profiling studies) cannot conclude how many RPFs exactly belong to actually translating ribosomes. This would require precise measurements of protein output from specific transcripts under at least two conditions, which is beyond the scope of the current work. We note, however, that none of our conclusions requires knowledge of the exact number of RPFs that belong to translating ribosomes.

I understand that the authors would prefer to keep this panel in order to reconcile the marked ribosome occupancy of the c-Myc transcript with the reduction in protein production, but the current data do not allow any solid claim on translational stalling. Therefore, I suggest to tune-down the ribosome pausing hypothesis, moving Figs 3F-G to supplementary data and acknowledging that these are just preliminary data. Please, also remove 'ribosome pausing' from the heading in lane 155 and modify the reading of this section accordingly. (By the way, I failed to see the precise nucleotide number of the 'stalled' peak indicated in the figure).

We also understand the reviewer's concern regarding the significance of the ribosome accumulation at a specific site in the c-myc transcript. We note that our observation of ribosome accumulation is based on 4 datasets (replicates of each compared condition). We realized that we did not report the p-values in our previous version and we have now corrected this omission. The p-value for the reported accumulation is $p = 0.023$, which is statistically significant, and thus unlikely to arise by chance." Position of the suspected "stalled" peak was also added. New version of the figure is presented below:

Given that the ribosome accumulation is statistically significant, we suggest that it is best to provide the panels in the main figures.

Of note, heading from lane 158-159 was updated as follows:

SCF^{Fbxo4}-dependent hnRNPk polyubiquitylation determines the rate of c-Myc protein synthesis via 5'UTR initiated translation and ribosome accumulation at ORF

On another note, as the authors recognize, regulation of IRES-mediated translation is independent to ribosomal stalling. Thereby the statements in lanes 191-192 '...suggests that SCFFbxo4-dependent regulation of hnRNPk activity contributes to pausing at these sites. To address the mechanism, we considered that SCFFbxo4 might regulate hnRNPk-dependent c-

Myc protein synthesis at its Internal Ribosome Entry Site (IRES)' are confusing. Re-writing this section to tune-down the ribosomal stalling part should solve this.

The fragment of the section between lane 186 – 196 has been re-phrased and language was tuned down as follows:

The increase of c-Myc RPFs in Fbxo4^{-/-} cells corresponds with observed higher protein output. Upon hnRNPk knock-down, the apparent translation efficiency (RPF/RNA ratio) increased (Fig 3F), while c-Myc protein level was reduced (Fig 3B). To understand this paradox, we analyzed the distribution of RPF reads across the c-Myc ORF. The higher apparent translation efficiency in Fbxo4^{-/-} cells indicates increased ribosomal occupancy and correlates with a robust increase in c-Myc protein synthesis upon Fbxo4 loss (Fig S3I). Interestingly, RPF distribution following hnRNPk loss revealed a transcript region where the RPF occupancy is elevated regardless of Fbxo4 status (Fig 3G; red arrows). These peaks identify sites of ribosome accumulation in the absence of hnRNPk. Since c-Myc synthesis declined, the peaks noted are consistent with ribosome pausing at these sites as a cause for the noted reduction in c-Myc levels and synthesis. From these data, we concluded that SCFFbxo4-dependent regulation of hnRNPk activity contributes to ribosome accumulation at these sites.

2) Table S1: The Table contains 439 genes, of which 53 are Myc targets. These numbers do not fit with the main text, where approx 700 genes are mentioned, of which 77 are Myc targets. Please, add the remaining genes to the Table or correct your statement.

Table S2 is an excel file containing two sheets with the list of up and downregulated genes. Indeed, the downregulated gene list contains 53 targets. However, the spreadsheets combined presents 77 c-Myc targets as we stated in the body text.

3) Figure S3L: The new Figure is much better, thank you. As a frequent user of PNK and CLIP methods, though, I usually see that high RNase concentrations lead to compression of the 32P-RNA smear above the size of the studied protein into a sharp band at the size of the studied protein, which is surprisingly absent in this figure. Probably the RNA is over-digested, but to really ensure that hnRNPk is present, and that the band indicated by the arrow actually corresponds to hnRNPk, the very same 32P-RNA membranes should be subjected to Western blot against myc-hnRNPk. The band of hnRNPk should exactly coincide with the arrow, and should also be present in the H lanes.

We appreciate the reviewers comment regarding high RNase concentrations. Indeed, this is the case. As proposed by the reviewer, the high RNase that we used cause the RNA to be degraded but did not affect our protein as expected. We agree that it is ideal that the very same 32P-RNA membrane be subjected to WB for myc-hnRNPk to show the intact myc-hnRNPk but it is not always doable as the radiation signal can be too strong, which in turn will mask the band. We therefore took another approach. In these experiments we routinely use two different concentrations of RNase (high and low) and the low RNase treated samples are further processed for sequencing if desired. As mentioned in the rebuttal letter and included in the scheme that was added to Figure S3L (shown below), cells were subjected to UV crosslinking, harvested, and hnRNPk was pulled down using a myc antibody. Next the samples were treated with different concentrations of RNase (high and

low) and split to either RNA radiolabeling procedure or (10% of the material) for immunoblotting to ensure equivalent IP of the hnRNPk protein after different RNase concentrations. We only included the input and the samples from the low RNase concentration in the figure (bottom right panel) for simplicity. As shown below, we now provide the high concentration as well to document intact hnRNPk. We detected hnRNPk both in Ub+ or Ub- in high RNase treated cells which implies that high RNase did not affect the protein. In this new version, we edited the scheme so it will be clear and added western-blot analysis from 10% of the IP sample subjected to high concentration of RNase.

Minor comments:

Lane 101: 'We noted that the RNA binding motifs for hnRNPk were highly enriched in mRNAs dysregulated upon Fbxo4 ablation (Fig 1F)'. It is more correct to say 'We noted that mRNAs dysregulated upon Fbxo4 ablation were highly enriched in hnRNPk motifs'.

We agree, expression as proposed by reviewer is better. The sentence has been replaced.

Figure 1C: Please, mention in the main text that oRNament analysis on targets responding at the level of RPF has also been run and show the same (3rd) position of hnRNPk in terms of motif abundance within KH domain RBPs (data not shown).

The main text has been modified as requested in the lane 103-105:

Of note, oRNament analysis on targets responding at the level of RPF has also been run and show the same (3rd) position of hnRNPk in terms of motif abundance within KH domain RBPs (data not shown).

Figure S1D: Please, mention in the figure legend or in main text that N-terminal truncations are expected to result in impaired substrate recognition because functional/active SCF-

Fbxo4 requires dimerization mediated by the N-terminal D-Domain.

The figure legend has been adjusted accordingly:

N-terminal truncations shows impaired substrate recognition due to lack of D-domain mediated Fbxo4 dimerization required for SCF-Fbxo4 activity.

Figure S5A: Can the authors, please, show quantification of several experiments? This would strengthen a reduction in hnRNPK-Fbxo4 interaction upon Fbxo4 mutation.

Three independent experiments have been quantified and presented as a fold change. Bolded number refers to displayed figure. A proper reference has been made by hashtag and explained in the figure legend.

Reviewer #2 (Remarks to the Author):

The authors satisfactory addressed all my concerns.

Reviewer #3 (Remarks to the Author):

In the revised manuscript, the authors have addressed most of the previous concerns. No further question.